# A monolithic immersion metalens for imaging solid-state quantum emitters

Tzu-Yung Huang[1,8], Richard R. Grote[1,5,8], Sander A. Mann[2,6], David A. Hopper [1,3], Annemarie L. Exarhos [1,7], Gerald G. Lopez[4], Amelia R. Klein [1], Erik C. Garnett [2] & Lee C. Bassett [1]

Quantum emitters such as the diamond nitrogen-vacancy (NV) center are the basis for a wide range of quantum technologies. However, refraction and reflections at material interfaces impede photon collection, and the emitters' atomic scale necessitates the use of free space optical measurement setups that prevent packaging of quantum devices. To overcome these limitations, we design and fabricate a metasurface composed of nanoscale diamond pillars that acts as an immersion lens to collect and collimate the emission of an individual NV center. The metalens exhibits a numerical aperture greater than 1.0, enabling efficient fiber-coupling of quantum emitters. This flexible design will lead to the miniaturization of quantum devices in a wide range of host materials and the development of metasurfaces that shape single-photon emission for coupling to optical cavities or route photons based on their quantum state.

[1] Quantum Engineering Laboratory, Department of Electrical and Systems Engineering, University of Pennsylvania, 200 S. 33rd Street, Philadelphia, PA 19104, USA. [2] Center for Nanophotonics, AMOLF, Science Park 104, 1098 XG Amsterdam, The Netherlands. [3] Department of Physics and Astronomy, University of Pennsylvania, 209 S. 33rd Street, Philadelphia, PA 19104, USA. [4] Singh Center for Nanotechnology, University of Pennsylvania, 3205 Walnut St., Philadelphia, PA 19104, USA. [5]Present address: Rockley Photonics Inc., 234 E. Colorado Blvd, Suite 600, Pasadena, CA 91101, USA. [6]Present address: Photonics Initiative, Advanced Science Research Center, City University of New York, New York, NY 10031, USA. [7]Present address: Department of Physics, Lafayette College, Easton, PA 18042, USA. [8]These authors contributed equally: Tzu-Yung Huang, Richard R. Grote. Correspondence and requests for materials should be addressed to L.C.B. (email: lbassett@seas.upenn.edu)

Solid-state quantum emitters have emerged as robust single-photon sources[1] and addressable spins[2]—key components in rapidly developing quantum technologies for nanoscale magnetometry[3], biological sensing[4], and quantum-information science[5]. Performance in these applications, be it magnetometer sensitivity[6] or quantum key generation rate[7], is limited by photon-collection efficiency. However, efficient collection of a quantum emitter's photoluminescence (PL) is challenging as its atomic scale necessitates diffraction-limited imaging with nanometer-precision alignment, oftentimes at cryogenic temperatures or in other situations incompatible with free-space bulk optics. Beyond their atomic scale, the challenges associated with coupling to solid-state quantum emitters are exacerbated by the high refractive index of their host substrates. Diamond, for example, has a refractive index of $n_D \sim 2.4$ at visible wavelengths, which traps photons in the material by the total internal reflection for propagation vectors oriented beyond $\theta_c \sim 25°$ from the surface normal of a planar air interface. Furthermore, imaging through more than a few microns of diamond with a high-numerical-aperture objective results in spherical aberrations that severely limit collection efficiency. While a number of nanophotonic structures have been investigated for increasing emission from diamond nitrogen-vacancy (NV) centers through Purcell enhancement[8–12], these devices require NV centers positioned close to diamond surfaces, which degrades their spin[13] and optical properties[14].

For this reason, a common approach to minimizing optical losses when addressing single NV centers in bulk diamond is to mill or etch a hemispherical surface, known as a solid immersion lens (SIL), around the NV center of interest[15]. By ensuring uniform optical path length and reflectance for rays emanating to all angles, SILs remove the losses caused by the total internal reflection and spherical aberration. SILs have enabled numerous advances in quantum optics using NV centers, including all-optical quantum control[16] and loophole-free violations of Bell's inequality[17]. However, a high-NA objective lens is still required to image a quantum emitter through a SIL. For quantum-optics experiments, a cryostat that can accommodate a vacuum-compatible objective and associated optomechanics must be used, or the optical losses associated with imaging through a cryostat window must be accepted. Neither option provides a clear route for packaging quantum emitters in a scalable fashion.

Since quantum emitters are point sources with relatively narrow emission spectra, the compound optical system of a microscope objective, designed for broadband imaging with a flat field-of-view, is not actually necessary for efficient photon collection. Flat optics, such as phase Fresnel lenses used to image trapped ions in ultra-high-vacuum cryostats[18], are an attractive alternative; however, a flat optic on its own cannot compensate for the high refractive index of a solid-state quantum emitter's host material. The ideal solution is a flat optic fabricated at the air/diamond interface to form a planar immersion lens; such a design can be realized using the concept of a metasurface.

Metasurfaces have recently gained attention as they offer design flexibility for optical components with arbitrary phase responses[19, 20]. In particular, diffractive optics[21, 22], high-contrast gratings[23, 24], and more recently, dielectric metalenses[21, 25–28] comprised of high-refractive-index dielectric elements such as $TiO_2$ and amorphous silicon have been demonstrated with high transmission efficiency and diffraction-limited focusing. While spherical and chromatic aberrations limit the field-of-view of single-element dielectric metalenses as compared with aberration-corrected multi-lens objectives[21], they are ideally suited for collimating emission from point sources[29]. When fabricated at a material interface, a metalens can be designed to use the underlying substrate as an immersion medium[25, 28, 30] to overcome the total internal reflection losses in a similar manner to a SIL; see Supplementary Note 1, Supplementary Fig. 1, and Supplementary Table 1.

Building on these advances, we leverage diamond's high refractive index to design and fabricate a 27.9-μm-diameter (19.3 μm effective aperture) metalens composed of subwavelength pillars etched into the surface of a single-crystal substrate that collimates the emission of an individual NV center located ~20 μm beneath the surface (Fig. 1a). The metalens eliminates the need for a collection objective by operating as an immersion lens with a numerical aperture (NA) greater than 1.0. This marks the first step in designing and fabricating metasurfaces for controlling photons from quantum emitters using only top-down fabrication techniques and provides a clear pathway to packaging quantum devices by eliminating the need for an objective.

## Results

**Immersion metalens design and fabrication.** The metalens is fabricated using standard, top-down, electron-beam lithography and $O_2$-based dry etching. Its pillars approximate a desired continuous

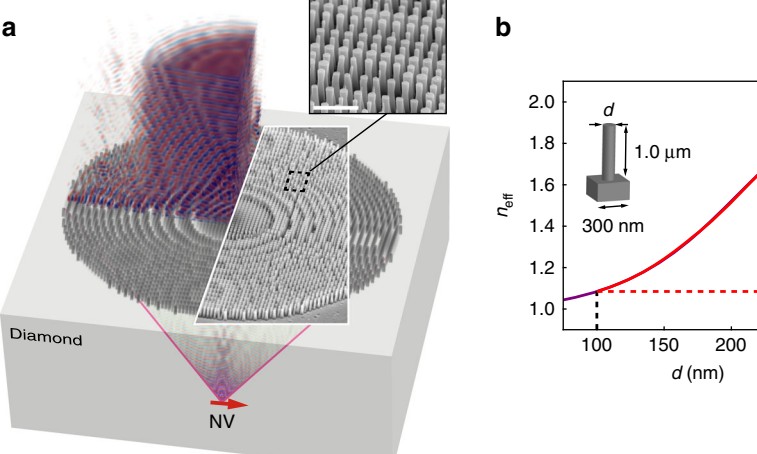

**Fig. 1** Diamond immersion metalens. **a** Subwavelength pillars extending from the surface of a single-crystal diamond substrate are designed to create a high-numerical-aperture immersion lens for coupling nitrogen-vacancy (NV)-center photoluminescence to a collimated beam in air. Inset: Scanning electron microscope (SEM) image of fabricated metalens with closeup of etched diamond pillars. The scale bar corresponds to 1 μm. **b** Bloch-mode effective index, $n_{eff}$, and corresponding optical pathlength difference, $\phi$, as a function of pillar diameter, $d$, at $\lambda = 700$ nm. This map is used to create the lens pattern shown in **a**

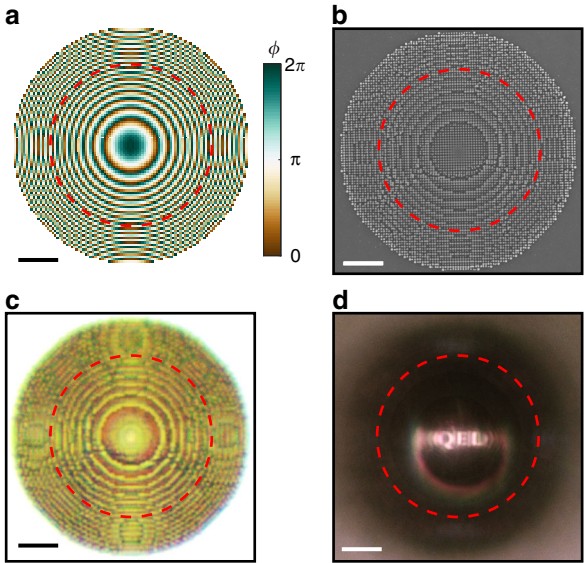

**Fig. 2** Metalens design and fabrication. Top–down images of: **a** the discretized Fresnel phase profile used for the design; **b** SEM image of the fabricated structure, **c** bright-field reflection optical image of the metalens surface; **d** image of a macroscopic chromium shadow mask with the Quantum Engineering Lab logo, ⟨Q|E|L⟩, formed through the metalens in a bright-field transmission microscope. Red, dashed lines indicate the effective aperture of the metalens. All scale bars denote 5 µm. Reflected light from the planar diamond surface surrounding the metalens in **c** saturated the camera, making it appear white

phase profile, $\phi(x, y)$, on a square grid by mapping the pillar diameter, $d$, to the effective refractive index, $n_{eff}$, of the lowest-order Bloch mode supported by the pillar (Fig. 1b). We use a Fresnel lens phase profile in conjunction with Fig. 1b to assign a pillar diameter to each grid point. The discretized phase profile for a focal length $f = 20$ µm at $\lambda = 700$ nm is shown in Fig. 2a, with a corresponding SEM image of the fabricated structure shown in Fig. 2b. Since the effective refractive index of each pillar is between the refractive index of air and the refractive index of diamond, the metalens is inherently anti-reflective, as evidenced by the bright-field reflection microscope image shown in Fig. 2c; see also the simulation and measured lower-bound on the metalens reflectance in Supplementary Fig. 13a. To demonstrate that the structure operates as a lens, in Fig. 2d we use a transmission microscope to form an image through the metalens of a chromium shadow mask illuminated from below the diamond; see Supplementary Note 4 and Supplementary Fig. 6 for details.

**Metalens performance and characterization**. We characterize the metalens using a combination of three-dimensional full-field electromagnetic simulations and confocal-scanning optical microscopy. Placing an NV center at the metalens focus in diamond results in collimation of the emitter's PL in air, as illustrated by the simulations in Fig. 3a. The PL can then be coupled into low-NA collection optics with high efficiency, as shown by the calculations in Fig. 3b. The black dot-dash line in Fig. 3b indicates the NA = 0.1 fiber used in our measurements, while the solid black line corresponds to the NA = 0.19 achromatic collection lens. Simulations of a plane wave launched from air and focused through the metalens were also performed; the simulated values of NA and focal length, $f$, shown as open squares in Fig. 3c and 3d, respectively, are extracted from the simulated electric field profile at each wavelength. The simulation for $\lambda = 700$ nm is shown in Fig. 3e. We use confocal-scanning optical microscopy in a double-pass geometry to produce high-resolution

scans of the metalens focal spot and its position inside the diamond (see the Methods section). These measurements are analyzed to obtain the experimental values of NA and $f$ shown in Fig. 3c, d (black points), demonstrating that the metalens has NA > 0.9 across wavelengths spanning the NV center's full emission spectrum, with NA = $1.10^{+0.12}_{-0.09}$ at $\lambda = 700$ nm. To enable this accurate comparison of simulation and measurement, we have numerically modeled the microscope's point-spread function using confocal measurements of isolated NV centers and deconvolved it from the focal-spot measurements to reveal the metalens's transverse and axial field profiles. Even with no free parameters, the deconvolved field profiles in Fig. 3f show excellent agreement with the simulations in Fig. 3e, as evidenced by the transverse and axial cross sections shown in Fig. 3g. Details regarding the characterization measurements, deconvolution analysis, and calculations of the transmission, collection, and focusing efficiency are available in the Methods, Supplementary Notes 5–6, and Supplementary Figs. 7–14.

**Imaging an NV center with the immersion metalens**. To image an NV center with the metalens, we focus a 532-nm pump beam through the backside of the substrate using an oil immersion objective (Fig. 4a). The confocal collection/excitation volume of the objective is axially positioned in the plane of the metalens focus and is rastered using a fast-steering mirror (FSM). NV-center PL at each scan position is simultaneously measured by two fiber-coupled single-photon counting modules (SPCMs): one is aligned to the metalens, and the other is aligned to the confocal path through the objective. The counts collected by the SPCMs at each point of the FSM raster scan form the images shown in Fig. 4b, c. The lenses in the metalens path (L1, L2 in Fig. 4a) re-collimate the diverging metalens output beam so that a 568-nm long-pass filter (LPF) can be inserted to block the pump beam.

Figure 4b and c both exhibit a bright spot at the same lateral position, denoted by the black dashed circles. We fix the FSM position at the center of this spot and measure the PL signals ($S_{ML}$, $S_{obj}$) through the metalens and objective paths, respectively. Background signals are separately recorded from a position off the spot but within the metalens field of view. The background-subtracted spectra of both paths (Fig. 4d) clearly exhibit the NV center's zero-phonon line at 637 nm and characteristic phonon side band. Background-subtracted PL saturation curves (Fig. 4e) display saturation count rates of $87.3 \pm 2.8$ photons/ms and $24.9 \pm 0.4$ photons/ms when measured through the metalens and objective, respectively. The objective NA is limited to 0.75 in order to mitigate spherical aberrations. Finally, we measure the second-order cross-correlation function, $g^{(2)}(\tau)$, between both paths. The background-corrected $g^{(2)}$ measurements (Fig. 4f) exhibit the characteristic antibunching dip and short-delay bunching of a single NV center with $g^{(2)}(0) = 0.175 \pm 0.031$, clearly demonstrating that the spots in Fig. 4b, c are indeed the same single-photon emitter. Details regarding the background-correction analysis are available in the Methods, Supplementary Note 7, and Supplementary Figs. 15, 16.

**Discussion**

The immersion metalens lays the foundation for future advances in controlling light-matter interactions for quantum emitters in high-refractive-index substrates. By integrating the typical objective/SIL combination onto the quantum emitter's host substrate, the metalens has the potential to enable direct fiber coupling of quantum emitters. In our experiment, two relay lenses and a free-space long-pass filter were used to prevent the pump beam from entering the collection fiber (L1, L2, and LPF in Fig. 4a). However, the metalens output can be coupled directly

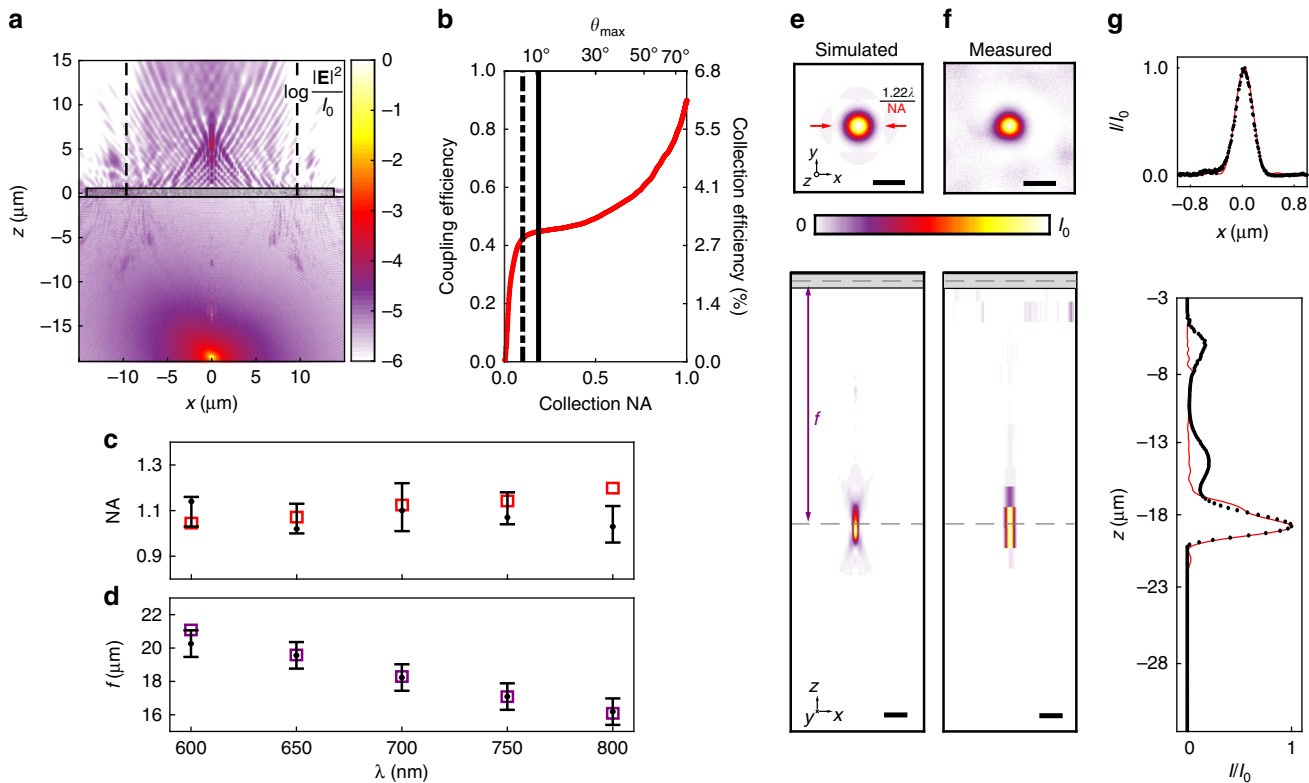

**Fig. 3** Metalens performance. **a** Simulated steady-state field intensity produced by the two optical dipoles of an NV center (with [111] orientation) placed at the metalens focus at $\lambda = 700$ nm. The position of the metalens is indicated by the gray box. Dashed black lines indicate the 19.3-μm effective aperture of the metalens. **b** Coupling efficiency and overall collection efficiency as a function of the acceptance angle of the collection optics placed after the metalens, calculated from the simulations shown in **a**. The dashed and solid lines indicate NA values 0.1 and 0.19, corresponding to the NA of the optical fiber and collimating lens used in the measurements, respectively. **c** Measurement (black points) and simulation (open red squares) of the metalens NA as a function of wavelength. Error bars represent the uncertainty in the objective NA, as described in the Methods section. **d** Measured (black points) and simulated (open purple squares) effective focal length. Error bars represent the uncertainty in estimating the positions of the focal spot and diamond surface. **e**, **f** Transverse (x–y, top) and axial (x–z, bottom) cross-sections of the metalens focal spot at $\lambda = 700$ nm: **e** simulated using a 3D finite-difference time-domain (FDTD) method, and **f** measured using a confocal scanning optical microscope with the microscope's point-spread function deconvolved. Gray boxes and dashed lines in axial cross-sections indicate the thickness of the metalens pillars and focus position, respectively. Scale bars denote 500 nm in transverse plots and 1 μm in axial cross-sections. **g** x (top) and z (bottom) line scans of the simulated (solid red curves) and measured (points) metalens focus at $\lambda = 700$ nm

into a fiber using a different excitation geometry or a commercially available multilayer-dielectric-coated fiber tip (available from Omega Optical, Inc., for example). Another limitation of our current demonstration is the inability to co-focus the pump beam and collection volume through the metalens due to chromatic aberration inherent to the Fresnel lens phase profile. Going forward, achromatic metalens designs[31, 32] can enable co-focusing of multiple wavelengths, or a second metalens can be incorporated on the backside of the diamond to focus the pump beam[33], replacing the objective in our experiment.

Unlike previous high-NA metalens demonstrations that relied on diffraction to focus wide angles far from the optical axis[22, 27], the high NA of our metalens is achieved by using diamond as an immersion medium. This implies that optimized design strategies could yield a diamond metalens with a substantially larger NA, potentially with a value approaching the maximum, $NA_{max} = n_D = 2.4$. Beyond lenses, the expanding body of research on metasurface design can be leveraged to explore phase profiles that shape emission from quantum emitter ensembles[34], compensate for an emitter's dipole orientation[35], control coupling to orbital–angular–momentum modes[24, 36, 37], and enable chiral quantum photonics[38]. An

immersion metasurface can also be incorporated with nanophotonic structures for Purcell enhancement, for example to collimate the output of a chirped grating structure[39, 40] or parabolic mirror[41] through the backside of the diamond, or to extend the cavity length of a fiber-based resonator cavity[42].

The immersion metalens promises major advances in performance and scalability of quantum devices. Its top-down fabrication processes are readily compatible with those used to fabricate on-chip microwave antennas and electric-field gates required for dynamic spin control and Stark shifting in quantum optics applications[16, 17]. Furthermore, the metalens design can be applied directly to other quantum-emitter systems, including spin defects in silicon carbide[43], quantum dots in III–V compound semiconductors[44], and rare-earth ions in laser crystals[45]. More generalized metasurface designs can mediate quantum entanglement[46] and interference[47] of quantum emitters. Ultimately, this demonstration has broad implications for nanophotonics, quantum optics, and quantum nanotechnology, as dielectric metasurface design will lead to compact, fiber-coupled single-photon sources, sensors, and quantum memories, with further potential applicability to designing diffractive optics for space[48] and Raman lasers[49].

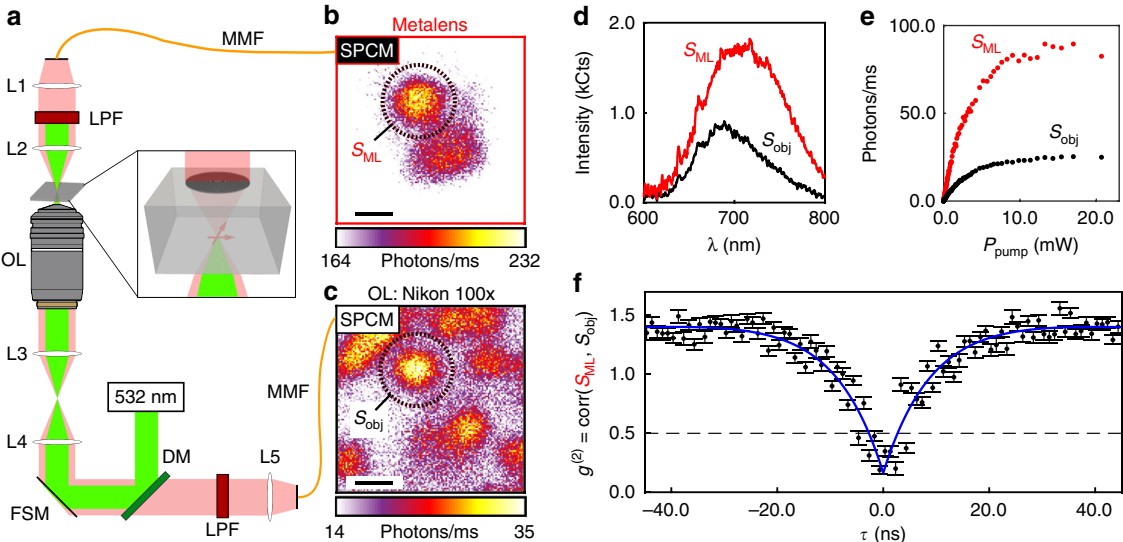

**Fig. 4** Imaging an individual NV center. **a** Experimental setup. MMF = multimode fiber, LPF = long-pass wavelength filter, OL = objective lens, FSM = fast-steering mirror, DM = dichroic mirror, SPCM = single-photon counting module, L1-L5 are achromatic lenses. **b** PL signal from the metalens when the 532-nm pump beam is rastered. **c** confocal PL image from the objective recorded simultaneously with **b**. Scale bars in **b** and **c** denote 500 nm **d**, PL spectra, and **e**, saturation curves of the metalens and objective signals, $S_{ML}$ and $S_{obj}$, corresponding to the spot circled in **b** and **c**, respectively. **f** Intensity cross-correlation between $S_{ML}$ and $S_{obj}$, confirming that the spot measured in both images is an individual NV center. The dashed line represents the single-emitter threshold. Measurements in **d**–**f** are background corrected. Error bars in **f** represent the Poisson uncertainty in each bin of the correlation function

## Methods

**Design**. The metalens was designed using the procedure devised by Lalanne et al. for TiO$_2$ deposited on glass[50]. The procedure was carried out as follows: first, the Bloch-mode effective index, $n_{eff}$, was calculated as a function of pillar diameter (Fig. 1b) on a subwavelength grid. The grid-pitch, $\Lambda$, was chosen to be just below the onset of first order diffraction, $\Lambda \leq \frac{\lambda}{n_D} = 291$ nm at $\lambda = 700$ nm, which was rounded up to $\Lambda = 300$ nm. The pillar height was chosen to be $h = 1.0$ μm and the minimum pillar diameter was set to $d_{min} = 100$ nm to ensure compatibility with our fabrication process. The maximum pillar diameter, $d_{max}$, was then found by determining the $n_{eff}$ required to achieve an optical pathlength increase of $2\pi$ relative to the minimum pillar diameter:

$$n_{eff}(d_{max}) = \frac{\lambda}{h} + n_{eff}(d_{min}). \tag{1}$$

The corresponding $d_{max}$ is found from the dispersion curve in Fig. 1b. The minimum and maximum pillar diameters are indicated in Fig. 1b (black dashed lines) along with the their relative optical pathlengths (red dashed lines).

The Fresnel phase profile in Fig. 2a was calculated by $\phi = n_D k_0 (f - \sqrt{f^2 + x^2 + y^2})$, with 93 grid points for a diameter of 27.9 μm measured by the grid edges at the maximum widths along the Cartesian design dimensions. The symmetry of this structure ensures polarization-independent focusing, which has been shown for similar designs using TiO$_2$ deposited on glass[51].

**Fabrication**. The metalens was fabricated on $3.0 \times 3.0 \times 0.15$ mm double-side-polished high-pressure/high-temperature (HPHT)-grown single-crystal diamond (Applied Diamond, Inc.). The diamond surface was cleaned in 90 °C Nano-Strip (a stabilized mixture of sulfuric acid and hydrogen peroxide, Cynaktec KMB 210034) for 30 min, followed by a 10 min plasma clean in a barrel asher with 40 sccm O$_2$ and 300 W RF power. The metalens pattern was proximity-effect corrected and written in hydrogen silsesquioxane (HSQ, Dow Corning, Fox-16) using a 50 keV electron beam lithography tool (Elionix, ELS-7500EX); see Supplementary Note 3, Supplementary Figs. 3–5, and Supplementary Table 2 for details. Prior to spin-coating HSQ, a 7 -nm adhesion layer of SiO$_2$ was deposited on the diamond surface by electron beam evaporation to promote adhesion. After exposure, the pattern was developed in a mixture of 200 mL of deionized water with 8 g of sodium chloride and 2 g of sodium hydroxide[52]. Our e-beam lithography process for HSQ on diamond can be found in ref. [53]. A reactive ion etch (RIE, Oxford Instruments, Plasma lab 80) was used to remove the SiO$_2$ adhesion layer and to transfer the HSQ pattern into the diamond surface. The SiO$_2$ adhesion layer was removed by a 1 min CF$_4$ reactive ion etch[54], followed by a 23 min O$_2$ RIE etch with a flow rate of 40 sccm, a chamber pressure of 75 mTorr, and an RF power of 200 W to form the diamond pillars. Finally, the HSQ hardmask was removed using buffered oxide etch.

**Simulations**. Calculations of $n_{eff}$, $\phi$ (Fig. 1b, left and right axes, respectively), and pillar transmission efficiency (see Supplementary Note 2 and Supplementary Fig. 2) were performed using 3D rigorous coupled-wave analysis (RCWA) based on the method developed by Rumpf[55]. The effective index of the pillars was calculated by solving for the eigenvalues of Maxwell's equations with the z-invariant refractive index profile of the pillar cross-section in a $300 \times 300$ nm square unit cell at $\lambda = 700$ nm. The eigenproblem was defined in a truncated planewave basis using $25 \times 25$ planewaves, with implicit periodic boundary conditions. Following these calculations, the pillar height was set to 1.0 μm with air above and homogeneous diamond below, and the complex amplitude transmission coefficient, $t$, of a normal incidence plane wave from air is calculated as a function of pillar diameter. The right axis of Fig. 1b was found by $\phi(d) = \angle t(d)$.

The focused spot in Fig. 3d was calculated using 3D finite-difference time-domain simulations (FDTD, Lumerical Solutions, Inc.). The 27.9 -μm-diameter metalens is contained in a $28.1 \times 28.1 \times 22.25$ μm total-field/scattered-field (TFSF) excitation source to reduce artifacts caused by launching a plane wave into a finite structure. Perfectly matched layers (PMLs) were used as boundary conditions 0.5 μm away from the TFSF source. The simulation mesh in the pillars was set to $10 \times 10 \times 10$ nm, increasing gradually to 50 nm along the propagation ($\hat{z}$)-direction into the diamond. Diamond is modeled with a non-dispersive refractive index, $n_D = 2.4$. An x-polarized planewave pulse ($\omega_0 \approx 2\pi \times 440$, $\Delta\omega \approx 2\pi \times 125$ THz) is launched from air toward the metalens surface. Steady-state spatial electric-field distributions, $\mathbf{E}(\mathbf{r})$, at five wavelengths ranging from 600 nm to 800 nm were stored, and the spatial fields at $\lambda = 700$ nm are plotted as transverse ($|\mathbf{E}(z = f)|^2$) and axial ($|\mathbf{E}(y = 0)|^2$) intensity distributions in Fig. 3e. The focal length, $f_{ML}$, at each wavelength (Fig. 3d) was determined by finding the grid point in the simulation cell where $|\mathbf{E}|^2$ is maximum. The spatial distribution of the steady-state field amplitude, $E_x(\mathbf{r})$, in Fig. 1a was simulated by removing the TFSF source and placing an $\hat{x}$-oriented dipole current source at the metalens focus position with a wavelength of 700 nm, ~18 μm below the metalens in diamond.

The electric-field intensity distribution in Fig. 3a was simulated in a similar manner by adding the intensity distributions resulting from two independent simulations of current sources with orientations corresponding to the two optical dipoles of the NV center[40]. This simulation was run using a $30.0 \times 30.0 \times 34.0$ μm cell with PML boundaries. The simulation mesh in the pillars was set to $30 \times 30 \times 25$ nm, increasing gradually to 50 nm along the z-direction into the diamond. The temporal pulse distribution used in the TFSF simulations was applied to the dipole sources, and steady-state spatial electric-field distributions, $\mathbf{E}(\mathbf{r})$, and dipole source power, $P_0$, were stored at five wavelengths ranging from 600 nm to 800 nm. These quantities were used to calculate the coupling efficiency plotted in Fig. 3b as described below.

Given a fixed acceptance angle of the collection optics following the metalens, $\theta_{collection} = \sin^{-1}(NA_{collection})$, the metalens coupling efficiency as a function of wavelength is defined as follows

$$\eta_{ML}(\theta_{collection}, \lambda) = \frac{\int_0^{\theta_{collection}} P_{air}(\theta_{air}, \lambda) d\theta_{air}}{P_D(\lambda)}, \tag{2}$$

where $P_{air}(\theta, \lambda)$ and $P_D(\lambda)$ are, respectively, the time-averaged powers transmitted through the metalens and emitted by a dipole current source located at the position of the metalens focal spot into the solid angle defined by the metalens NA. A 2D spatial

Fourier transform is performed at each wavelength on the FDTD-calculated transverse electric-field amplitudes in air 215 nm above the top surface of the metalens using the MATLAB function **fft2** with zero padding to increase the simulation cell size to $4097 \cdot 30 \times 4097 \cdot 30$ nm to calculate $P_{air}(\theta, \lambda)$, which is then integrated from normal incidence up to $\theta_{collection}$. $P_D(\lambda)$ is calculated from the geometrical efficiency[56] of the metalens (see Supplementary Note 6), and the total time-averaged power emitted by the dipole, $P_0(\lambda)$: $P_D(\lambda) = \eta_{geom}(NA_{ML}) \cdot P_0(\lambda)$. The spectrally averaged coupling efficiency plotted in Fig. 3b is calculated using a weighted sum over the NV center's spectrum as $\langle \eta_{ML} \rangle_\lambda = \int W(\lambda) \cdot \eta_{ML}(\lambda) d\lambda$, where $W(\lambda)$ is the spectrum measured through the oil immersion objective shown in Fig. 4d normalized such that $\int W(\lambda) d\lambda = 1$.

**Experimental.** Measurements of the metalens were carried out with a custom-built confocal microscope, comprised of an oil immersion objective with adjustable iris (Nikon Plan Fluor x100/0.5-1.30) and an inverted optical microscope (Nikon Eclipse TE200) with a $\hat{z}$-axis piezo stage (Thorlabs MZS500-E) as well as a scanning stage for the $\hat{x}$ and $\hat{y}$ axes (Thorlabs MLS203-1). The diamond host substrate was fixed to a microscope coverslip (Fisher Scientific 12-548-C) using immersion oil (Nikon type N) with the patterned surface facing upwards. A 30 mm cage system and SM1-thread components (Thorlabs) were used to create a fiber-coupled optical path above the stage of the inverted microscope. This configuration allowed for simultaneous excitation and measurement of the metalens from air (fiber-coupled path) or through the diamond (objective path). The objective path was routed outside the microscope body so that laser-scanning confocal excitation and collection optics could be added. A 4f relay-lens-system consisting of two achromatic doublet lenses (Newport, 25.4 mm × 150 mm focal length, PAC058AR.14) was used to align the back aperture of the objective to a fast-steering mirror (FSM, Optics in motion, OIM101), which was used to raster the diffraction-limited confocal volume in the transverse $x-y$ plane of the objective space. A 560-nm long-pass dichroic mirror (Semrock, BrightLine FF560-FDi01) placed after the FSM was used to couple a 532-nm excitation laser (Coherent, Compass 315M-150) into the objective, while wavelengths above 560 nm pass through the dichroic mirror and are focused into a 25-μm core, 0.1 NA, multimode fiber (Thorlabs M67L01) that can be connected to a single-photon counting module (Excelitas, SPCM-AQRH-14-FC) or a spectrometer (Princeton Instruments IsoPlane-160, 750-nm blaze wavelength with 1200 G/mm) with a thermoelectrically cooled CCD (Princeton Instruments PIXIS 100BX). Computer control of the FSM and counting the electrical output of the SPCM was achieved using a data acquisition card (DAQ, National Instruments PCIe-6323).

For the characterization measurements presented in Fig. 3, a broadband supercontinuum source (Fianium WhiteLase SC400) was coupled through a single-mode fiber (Thorlabs P1-630AR-2), collimated, and brought into the collection path of our microscope via a beamsplitter (Thorlabs BS014). A $f = 2.0$-mm collimating lens (Thorlabs CFC-2X-A) and a $f = 15$-mm achromatic lens (Thorlabs AC064-015-B) were used to couple the metalens to a single-mode fiber (Thorlabs P1-630A-FC-1) and a fiber retroreflector (Thorlabs P1-630R-P01-1) which, upon reflection, recreates a 28-μm-diameter Gaussian beam that emulates the planewave source used in our FDTD simulations. The excitation wavelength is set by passing the supercontinuum beam through a set of linearly variable short-pass (Delta Optical Thin Film, LF102474) and long-pass filters (Delta Optical Thin Film LF102475) prior to fiber-coupling, which can be adjusted to filter out wavelengths with <8-nm bandwidth or be removed completely for broadband excitation. The transverse profile and cross-sections in Fig. 3f, g were measured by filtering the supercontinuum source to a single wavelength and rastering the FSM while collecting counts in the SPCM connected to the confocal path at each scan position. This process is repeated for a series of z-stage positions to measure the axial profile, which is shown in Fig. 3f at $\lambda = 700$ nm. These data were used to find the metalens focal length as a function of wavelength in Fig. 3d.

In Fig. 4, the fiber-coupled path was used to image a single NV center through the metalens, as shown in Fig. 4a. This was achieved with two achromatic doublet lenses (L1 and L2) with focal lengths of $f = 13$ mm and $f = 15$ mm (Thorlabs AC064-013/015-B), respectively, aligned to a 25-μm-core, 0.1 NA, multimode fiber (Thorlabs M67L01). The multimode fiber was then connected to a second SPCM (Excelitas, SPCM-AQRH-14-FC), allowing for simultaneous PL collection from both the fiber-coupled and objective paths while scanning the excitation source. The long-pass filters (LPF) in both collection lines consisted of 532-nm and 568-nm long-pass filters (Semrock, EdgeBasic BLP01-532R, EdgeBasic BLP01-568R) for spectra measurements, with an additional 650-nm long-pass filter (Thorlabs, FEL0650) in both paths to improve the signal-to-background for PL, saturation, and cross-correlation measurements. The outputs of both SPCMs were connected to a time-correlated single-photon counting card (TCSPC, PicoQuant, PicoHarp 300) to collect photon arrival-time data that were used to calculate cross-correlation functions (Fig. 4f). A full diagram describing the experimental setups for Figs. 3 and 4 can be found in Supplementary Fig. 7. Background spectra and saturation curves were measured at a transverse scan position away from the NV center, but still within the field-of-view of the metalens, and were subtracted from measurements taken on the NV center. This process was also used to determine the background for correcting cross-correlation data by interleaving 40 measurements off the NV center with 40 measurements taken on the NV center, each with a 5 min acquisition time. Further details on background-subtraction of the measurements in Fig. 4 are given in Supplementary Note 7.

**Analysis.** The NA of the metalens, $NA_{ML}$, plotted in Fig. 3c is calculated by fitting the simulated (open squares) and measured (black points) transverse focus spot at each wavelength (Fig. 3e, f, Supplementary Figs. 10 and 11) to the paraxial point-spread function of an ideal lens, an Airy disk[57],

$$ I = \left| \frac{2J_1(NA_{ML}k_0 r)}{NA_{ML}k_0 r} \right|^2, \tag{3} $$

where $k_0 = 2\pi/\lambda$ is the free space wavenumber and $r = \sqrt{x^2 + y^2}$ is the radial coordinate in the focal plane. Fits are performed using non-linear least-squares curve fitting (MATLAB function **lsqcurvefit**). The measured transverse focus spots used for the fits are obtained by an iterative deconvolution of the experimental profiles using the point-spread function of the microscope objective. The point-spread function is numerically calculated from confocal measurements of a single emitter; see Supplementary Note 5 and Supplementary Fig. 8. The deconvolution operations are performed using the Lucy–Richardson method (MATLAB function **deconvlucy**) with the number of iterations selected by minimizing the mean-squared-error (MSE) of the deconvolution algorithm. The MSE is calculated between the measured data and the re-convolution of deconvolved measured data with the objective point-spread function (MATLAB function **immse** and **convnfft**, respectively). The fit uncertainty is dominated by the uncertainty in the objective's NA, which was verified to be $0.76 \pm 0.03$. The confidence intervals plotted as error bars in Fig. 3c reflect this range of objective NA values used as input for the deconvolution and fitting analysis.

The entrance pupil, $D$, of the metalens can be calculated by geometry using $NA_{ML}$ and $f_{ML}$:

$$ D = 2f_{ML}(\lambda) \tan\left[\sin^{-1}\left(\frac{NA_{ML}}{n_D}\right)\right]. \tag{4} $$

Using Fig. 3e along with Eq. (4), we find that $D = 19.3$ μm, which is smaller than the physical 27.9-μm diameter of the metalens. This indicates a maximum collection angle inside the diamond of $\theta_{max} = \sin^{-1}\left(\frac{NA_{ML}}{n_D}\right) = 27.8°$. Despite this limited collection angle, Fig. 3c clearly illustrates $NA_{ML} > 1.0$, which can be increased by using diffractive designs for larger angles.

The focal length of the metalens in Fig. 3d was determined by measuring the distance between the metalens surface and the focused spot formed below the metalens using the piezo stage of the microscope. The distance traversed by the piezo stage is then scaled by a factor of $\approx \frac{n_D}{n_{oil}}$ to compensate for distortions caused by imaging through diamond[58]. Further details are given in Supplementary Fig. 9. Corrections for the PL background in Fig. 4d–f were performed by recording background levels near the NV center of interest. The NV center's PL saturation curve is nonmonotonic due to ionization and recombination, together with shelving in the spin-singlet manifold. At high powers approaching saturation, saturation curves cannot be fit with the typical two-level saturation model[40]. For the measurements in Fig. 4e, the PL has clearly saturated for $P_{pump} > 12$ mW; therefore, we calculate the saturated count rate and corresponding uncertainty from the average and standard deviation of the data recorded above this power level. Background curves are shown in Supplementary Fig. 15.

Background correction of the cross-correlation data in Fig. 4f was performed using the following relationship[59]:

$$ g_{bc}^{(2)}(\tau) = \frac{g^{(2)}(\tau) - (1 - \rho^2)}{\rho^2}, \tag{5} $$

where $g^{(2)}(\tau)$ is the measured second-order correlation function and $\rho = 0.26 \pm 0.01$ is the ratio of background-corrected signal to total signal determined by 40 repeated measurements. After background correction, $g_{bc}^{(2)}(\tau)$ is fit with the following analytical function:

$$ g_{bc}^{(2)}(\tau) = 1 - Ae^{-\frac{|t-t_0|}{\tau_1}} + Be^{-\frac{|t-t_0|}{\tau_2}}, \tag{6} $$

which corresponds to the the approximation of the NV center as a 3-level emitter[60]. The fit coefficients are as follows: $A = 1.31 \pm 0.03$, $B = 0.48 \pm 0.01$, $t_0 = -2.2 \pm 0.2$ ns, $\tau_1 = 8.8 \pm 0.3$ ns, $\tau_2 = 221 \pm 6$ ns. The best-fit value of $g^{(2)}(0) = 0.17 \pm 0.03$. Background-uncorrected data and additional details are given in Supplementary Note 7 and Supplementary Fig. 16.

## Data availability
The data that support the findings of this study are available from the corresponding author upon reasonable request.

## Code availability
Codes used for the calculation of effective refractive index as a function of pillar diameter, the mapping to a discretized phase profile, and the generation of GDS layout files are available at https://github.com/penn-qel/metalens-fresnel.

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

## Acknowledgements

This work was supported by the National Science Foundation (NSF) under awards ECCS-1553511 and ECCS-1842655. The authors gratefully acknowledge use of facilities and instrumentation supported by NSF through the University of Pennsylvania Materials Research Science and Engineering Center (MRSEC) (DMR-1720530). The authors also acknowledge support from the University Research Foundation and the Singh Center for Nanotechnology at the University of Pennsylvania, a member of the National Nano-technology Coordinated Infrastructure (NNCI), which is supported by the NSF (Grant ECCS-1542153). S.A.M. and E.C.G. were supported by the Netherlands Organisation for Scientific Research (NWO) under the European Unions Seventh Framework Programme ((FP/2007-2013)/ERC grant agreement no. 337328, Nano-EnabledPV). The authors thank Meredith Metzler for assistance in developing the $O_2$-based reactive ion etch.

## Author contributions

T.-Y.H, R.R.G., S.A.M., A.R.K., and E.C.G. performed the design and simulations; R.R.G. and G.G.L. fabricated the metalens; T.-Y.H., R.R.G., D.A.H., A.L.E., and L.C.B. performed the measurements and analysis. All authors contributed to writing the paper.

## Additional information

**Competing interests:** The authors declare no competing interests.

