## [Peer Review File · Nature Communications]

Reviewers' comments:

Reviewer #1 (Remarks to the Author):

The manuscript presents design, fabrication, and characterization of a small diffractive lens that is etched directly into the surface of a diamond substrate. The lens is relatively small (15-micron radius) and is designed to focus 700nm incident light inside the substrate. The lens is composed of diamond pillars with different cross-section dimensions that are arranged on a lattice with the period of 300nm. The pillars operate in the effective index regime (i.e., almost no resonance enhancement) and referring to the lens as a metalens is not strictly correct. In fact, the lens operation is similar to the following graded index lens previously reported:

West, Paul R., et al. "All-dielectric subwavelength metasurface focusing lens." *Optics Express* 22.21 (2014): 26212-26221,

which is also monolithic immersion lens made with subwavelength pillars. The main difference with the current manuscript is the material (diamond vs silicon), and the motivation/application. The concept of immersion metalenses for focusing inside high refractive index materials is not new, for example, see:

Ho, John S., et al. "Planar immersion lens with metasurfaces." *Physical Review B* 91.12 (2015): 125145.

and more recently a TiO₂ immersion metalens with the NA of ~1.1 has been reported:

Chen, Wei Ting, et al. "Immersion meta-lenses at visible wavelengths for nanoscale imaging." *Nano letters* 17.5 (2017): 3188-3194.

and imaging with the resolution of ~200nm has been demonstrated. Unfortunately, these works have not been cited in the manuscript. In addition, the manuscript has several technical issues:

1. The period of the pillar lattice is too large for proper operation of the lens. To avoid excitation of higher order diffraction orders, the lattice period should be smaller than $\text{wavelength}/(n \cdot (1 + \sin(\theta)))$ where n is the refractive index of the substrate and θ is the deflection angle. The deflection angle at the edge of the lens reported in the manuscript is ~35 degrees and the period of the lattice should be smaller than 183 nm. However, the period selected by the authors is 300nm which causes excitation of higher order diffraction orders even close to the center of the lens and reduces the lens efficiency. This effect can be seen in the simulation result shown in Fig. 1a which shows only a small portion of the lens close to the center actually collimates the light emitted by the dipole source.

2. The main advantage of metalenses over conventional diffractive lenses, which are typically fabricated in glass and polymers, is their ability in achieving high NA focusing with high efficiency. Although the current manuscript claims (see 2 below) a high numerical aperture for the lens, it does not discuss/characterize the focusing efficiency of the lens. High NA could be readily achieved by

patterning a simple Fresnel zone plate on the diamond surface, the main point of using a binary blazed structure is achieving higher efficiency, which has not been reported in the manuscript. The simulated and measured focusing efficiencies (the ratio of the power in the focal spots shown in Figs. 3a and 3c to the optical power incident on the lens aperture) should be reported.

3. The characterization of the focal spot of the lens is not conclusive. Although the simulation result predicts a small focal spot, the phase errors due to fabrication might lead to a larger focal spot. The actual measured focal spot of the lens is significantly larger than the predicted one. The authors attribute this issue to the aberrations of the microscope objective used to measure the spot, which might be correct. However, comparing the measured result with the convolution of the simulated spot and the PSF of the objective does not provide evidence of the small size of the focal spot. The convolution operation filters out high-frequency components and similarity between two filtered spots does not mean that the original spots were of similar size. The authors can test this by convolving a 5nm and a 700nm spots with the PSF of the microscope and comparing it with the measured spot.

The authors also have studied the saturation power of an NV center close to the focal spot to offer further evidence of focal spot size. However, the result of this characterization is also unreliable because of the following two issues. First, very large backgrounds (more than 80 % of the collected signal power) have been subtracted from the signals captured from the NV using the objective lens and the metalens. As a result, a small error in the estimation of the background may shew the measurement results significantly. Second, the effect of the polarization/orientation of the NV center has not been considered. The NV might be emitting more power toward the metalens than the objective.

Overall, the manuscript presents a potentially interesting application of metalenses, but do not offer much of a novel concept nor technique. The lens has not also been properly designed (lattice constant is significantly larger than what it should be), is most probably is less than 50% efficient, and the characterization results are not conclusive.

Reviewer #2 (Remarks to the Author):

Since this is a transparent review, I should note that I was an author of several papers on diamond Solid Immersion Lenses (SILs) fabricated using Focussed Ion Beam (FIB), which is a potentially competing technique the authors compare their work to. I have tried my best to weigh the pros and cons of the two techniques fairly.

The authors present a proof of principle demonstration of a dielectric metalens (ML), designed, and fabricated into the surface of diamond though top down clean room processes for efficient and in-

principle objective-less collection of fluorescence from diamond colour centres. They claim that the ML exhibits high transmission efficiency and outperforms a traditional free-space objective. They also suggest the structure is good for minaturization, on chip electronics, coupling to optical cavities, and routing of photons

I believe the paper represents an interesting application of dielectric ML, and is a novel demonstration in diamond. The fact that it is implemented using standard Ebeam masked dry etching diamond clean-room processing techniques is a definite plus point - in comparison to for example FIB fabrication of solid immersion lens SILs, in which each structure is made individually, making scalability more difficult. The authors present a pleasing amount of detail, with a pretty decent literature review on the design and fabrication processes, and the analysis appears to be valid (subject to a few comments or clarifications detailed below) - so I do feel that other researchers should be able to repeat the work - and if such structures become well used, then this will likely become a good reference resource.

I do have several concerns regarding the claim that the structures outperform traditional free space objectives, which I feel the authors should either clarify - or potentially tone down the claim.

Part of the concern with this claim is that it relies partly on the comparison of numerical aperture ($NA = n \sin \theta$, where n is the immersion material, and θ is the collection cone half angle) between the ML, and the 'traditional objective' - which may not be the ideal way to compare the two systems since it obscures several important details. The ML appears to show a numerical aperture exceeding 1, up to almost 1.2, which at first look is better than a traditional air objective limited by 1 NA, and comparable to standard oil objectives, typically up to 1.4.

The authors appear to use a oil immersion objective with adjustable NA from 0.5-1.3, although in the experimental details they appear to use an NA of 0.75 - which suggests they are using the objective at only ~60% of it's achievable NA. If this is the case it is not necessarily surprising that the ML outperforms it. In any case the authors should be much more explicit about what they are comparing the NA of the ML to.

The repeated claim of a NA larger than 1 including in the abstract is also potentially slightly misleading, since as the authors note, because the immersion material is diamond, the maximum achievable NA is 2.42, so in reality the performance of the ML in terms of it's light collecting power is comparable to an NA scaled by the diamond material of ~0.4-0.5. This means that although it will outperform a 0.9 NA air objective collecting from diamond (reduced to 0.4 NA by the diamond refractive index), it will likely not outperform a 1.4 NA oil immersion objective collecting from diamond (reduced to 0.6 NA), and certainly will not outperform either of these objectives in

combination with a diamond SIL (which eliminates the NA reduction). I would suggest that the inclusion of either estimated collection efficiencies, the NA scaled by the immersion material, or collection cone half angles would make the comparison more easy to understand, and therefore more valid (although once this is done it might be necessary to temper or remove the claim).

In terms of the claims to minaturization, and compatibility to on chip antennas as compared to SILs, I feel like the authors need to be a little more explicit as to the limits of the production of these ML. I note that the lens they present here has a diameter of 28 μm , for an NV 20 μm below the surface. This is a factor of almost 10 larger than the typical NV/SIL combination which has been used before - which then means that the coupling strength of surface mounted antennas are much less - and the danger of cross talk between NV centres is potentially much greater. Do the authors believe that the ML can be pushed to NV centres closer to the surface - allowing smaller structures and closer electronics - or is there a limit based on the achievable size of the diamond pillars which make it up.

Do the authors have any data on how alignment of the ML to the emitter effects their performance? Also comments on the field of view might be interesting.

It would also be interesting to comment on the scalability of the ML - for multiple NVs on a single chip. For example if several NV centres were positioned under one ML, or several NVs each under it's own ML, how easy is it to separate the collected photons from each NV. Presumably (?) the field of view for the 'objective' achromatic lens which replaces the microscope objective becomes much larger - which could make it possible to see more NV centres on one chip - however, angular separation might become more challenging.

Ultimately it would be good to dispassionately discuss whether the performance of such a ML can match that of a SIL with a high NA objective - how far can the NA be pushed towards 2.42..

In conclusion I believe this is a good paper, demonstrating a novel technique in diamond, with lots of useful detail on the techniques. I have some reservations on the claims made about the relative performance of the ML, and on the outlook for how the ML will influence the field - however assuming the authors clarify and/or reduce these claims I think it is interesting enough for Nature Communications.

In addition to this concern I have a few more specific comments on the text.

P1 Para1: 'number of photons detected.' small point, but I would say the collection efficiency is maybe better.

P1, Para3: 'Purcell enhancement' - it might be worth including a reference to the tunable external cavity style structures of for example Jason Smith, David Hunger, etc. (one example might be "Cavity-Enhanced Single-Photon Source Based on the Silicon-Vacancy Center in Diamond")

P2 Para2: 'narrow wavelength range' how narrow - the nv center has a spectrum over several hundred nm (600nm-800nm), while the commonly used excitation wavelength is at 532nm.

Fig 2: The scale bar here is 5um, while in the supplementary information it looks like the same mask is 10um. Is this an error - or is a field of view effect?

P3 para2: 'We characterize the metalens.....'. A semantic point, but personally I feel it is good to separate experimental characterization of real structures, with simulation of theoretical structures - perhaps you could make this clearer.

P3 para2: 'we numerically model the microscopes PSF'. Perhaps you could be clearer how you model the PSF. How does the numerical PSF compare with an experimentally determined PSF. Colour scale bars might be useful here too.

P3 para3: 'low reflectivity seen qualitatively'. I may be missing something, but I'm not sure how I can qualitatively see anything like this in 2c - it potentially depends a lot on what the contrast is.

P3 para3: 'low reflectivity..... to be below 12%'. Perhaps the authors should comment and compare with the expected fresnell reflectivity of a non structured diamond/air interface. Do the pillars or the fabrication in general add any extra scattering.

Fig 3: a,b, c, It might be good to add colour scale bars. Also to make it clear what the grey bar at the top of the x,z graphs are - presumably this is the pillar length, but good to make it explicit. f: the ripples are suggested to be because of ghosting from an optical element - but I can't see a corresponding characterisation of this effect as promised in the supplemental information. Without this could it be argued that the ripples come from some sort of self interference in the diamond structure?

Figure 4: b,c Colour scale bars! f: Would it be possible to make explicit how far $g_2(0)$ goes down. Also - in order for the reader to see how good/bad the background levels are it is nice to add a second axis on the right with the absolute correlations. The left hand scale bar looks a little odd.

P8 Para 4. 'Assuming that the excitation and collection paths have similar transmission efficiencies...' since this is an attempted calculation of the collection efficiency it would be good to explicitly look at the transmission efficiencies. I note that the lenses used in the 4f system are rated with AR coatings from 400-700nm. In my experience there can be much worse performance outside this range - which might end up causing poor performance in the confocal arm (where there are at least 3 extra optical elements). It should be possible to characterize or at least estimate the performance of each arm separately.

Reviewer #3 (Remarks to the Author):

This is an interesting paper presenting an alternative approach to the collection of PL from single NV centres buried inside diamond. The use of a metalens fabricated inside the diamond as an alternative to a SIL is certainly an idea worthy of exploration.

The paper is very well written and very clear.

However, the results presented fall somewhat short of what was promised in the abstract. Figure 4 certainly shows that the metalens works to collect the light from an NV centre, but this is on the back of the centre being illuminated by the output of an oil immersion objective. Figure 4 would be more interesting if it demonstrated that the ML can actually eliminate the need for an objective lens. Although the increase in the number of collected photons is impressive, I wonder if optimization of the traditional objective arrangement could not have yielded a similar result. Can the experiment be repeated so that the metalens provides the excitation and collection just as the objective lens does in figure 4?

A minor point relates to figure 3f which shows a reflectivity of around 12% or so. It would be helpful to compare this to the measured reflectivity of

Itemized responses to the March 29th reviewers' comments appear below. Comments by the editor and reviewers are formatted in blue text. Responses to each comment are formatted in normal font, along with a summary of the corresponding changes to the revised manuscript.

Editor Comments:

Your manuscript entitled "A Monolithic Immersion Metalens for Imaging Solid-State Quantum Emitters" has now been seen by 3 referees, whose comments are appended below. You will see from their comments copied below that while reviewer #2 and #3 they find your work of considerable potential interest, all three reviewers have raised quite substantial concerns that must be addressed. In light of these comments, we cannot accept the manuscript for publication in the present form, but would be interested in considering a revised version that addresses these serious concerns.

In particular reviewer #1 brings up novelty and characterization concerns. As there have been previous work on similar lens operation, we need your work to showcase the application and advantage in diamond. Following on from this the concerns of #2 regarding the claimed NA and the imaging of NV centres should be carefully answered. Please consider including more data to support your application demonstration.

Our Response: We appreciate your insight regarding the reviewers' comments and the importance of showcasing the novelty and impact of our work. We have performed additional measurements, simulations, and manuscript revisions to address these comments and to better illustrate the novelty and impact of our demonstration.

In particular, reviewer #1's comments encouraged us to more clearly emphasize that our manuscript represents the first use of a high-NA immersion metalens for imaging a quantum emitter. We have further emphasized the significance of our demonstration to showcase that immersion metalenses can be used to fiber-couple quantum emitters embedded in high-refractive index substrates, potentially with efficiencies comparable to or exceeding a high-numerical-aperture free-space objective used with a solid immersion lens, while being significantly less bulky, easier to manufacture, and more suitable for incorporation in a cryostat. In short, our demonstration is of considerable technological relevance for developing quantum devices and opens new possibilities for scientific discovery.

To address the reviewers' concerns regarding characterization of the device's optical performance, we have devised a method for obtaining higher-resolution measurements of the

metalens focal spot by incoherent imaging. **The improved accuracy of this new method has allowed us to obtain an experimental fit of the metalens NA = $1.073_{-0.038}^{+0.002}$ at 700 nm.** We have also added new measurements and simulations of the metalens focusing efficiency, coupling efficiency, and an expanded discussion of the metalens imaging performance in comparison to current state-of-the-art quantum-emitter imaging techniques. In the revised manuscript, we have incorporated this additional material into the manuscript to address the reviewers' concerns as thoroughly and convincingly as possible while conveying to a more general audience the significance and impact of our result.

Actions Taken:

- 1.** We have performed simulations of focusing efficiency vs. wavelength (Fig. S12c in the supporting information). The simulations are consistent with measurements of the metalens NA as described below, and confirm that the effective aperture of our structure, $D \sim 20 \mu\text{m}$, is smaller than the physical size of the structure. The focusing efficiency has been compared to other high-NA metalenses from the literature to illustrate that our design has similar performance, and a simulation of a zone plate with a comparable diameter to our metalens (but with far poorer performance) was performed to alleviate the concerns of reviewer 1.
- 2.** Simulations of the metalens coupling efficiency have been performed for dipole radiation patterns matching the physical orientation of the NV center's dipole axes in the diamond crystal (see Fig. 3a). Fourier analysis of the simulated metalens output fields have been performed to quantify the beam properties and the coupling efficiency as a function of collection-fiber NA (Fig. 3b). The coupling efficiency simulations are consistent with experimental estimates based on calibrated measurements of the saturated count rate from a single NV center imaged using the metalens.
- 3.** Incoherent imaging¹ was used to obtain higher resolution measurements of the metalens focal field profile. Together with an iterative deconvolution process to obtain the maximum likelihood estimate for the field profile, these measurements allow us to perform a numerical fit to the data to quantify the metalens NA as a function of wavelength. These new measurements are included in Fig. 3 of the main text and Fig. 10, 11 of the supporting information.
- 4.** To address the comments of Reviewer 2, we have added a section in the SI (Sec I) discussing the performance tradeoffs and key comparisons between metalenses and solid immersion lenses.

Reviewer 1:

The manuscript presents design, fabrication, and characterization of a small diffractive lens that is etched directly into the surface of a diamond substrate. The lens is relatively small (15-micron radius) and is designed to focus 700nm incident light inside the substrate. The lens is composed of diamond pillars with different cross-section dimensions that are arranged on a lattice with the period of 300nm. The pillars operate in the effective index regime (i.e., almost no resonance enhancement) and referring to the lens as a metalens is not strictly correct. In fact, the lens operation is similar to the following graded index lens previously reported: West, Paul R., et al. "All-dielectric subwavelength metasurface focusing lens." *Optics Express* 22.21 (2014): 26212-26221, which is also monolithic immersion lens made with subwavelength pillars. The main difference with the current manuscript is the material (diamond vs silicon), and the motivation/application. The concept of immersion metalenses for focusing inside high refractive index materials is not new, for example, see: Ho, John S., et al. "Planar immersion lens with metasurfaces." *Physical Review B* 91.12 (2015): 125145. and more recently a TiO₂ immersion metalens with the NA of ~1.1 has been reported: Chen, Wei Ting, et al. "Immersion meta-lenses at visible wavelengths for nanoscale imaging." *Nano letters* 17.5 (2017): 3188-3194. and imaging with the resolution of ~200nm has been demonstrated. Unfortunately, these works have not been cited in the manuscript.

Our Response: The reviewer is correct in asserting that our lens does not rely on resonant elements; however, the diamond pillars are on a subwavelength grid and can be treated as a locally homogenous effective medium with a continuously graded refractive index. This is the widely accepted definition of a metalens, as has been established with similar structures that operate in this regime (see, for instance, Ref. 52 of the main text). While older works have referred to the same structure as blazed-binary diffractive lenses [refs. 21,22,51 of the main text] or high-contrast gratings [refs 23,24 of the main text], we included these references to put this previous work in context. Even the paper by West *et al.* referenced by the reviewer as having similar operation to our lens is entitled, "All-dielectric subwavelength metasurface focusing lens." Semantic discussions aside, what really matters is that appropriate credit is given to relevant work predating the metalens era, such as refs [20-24] of the main text.

We agree that our metalens design is very similar to the work of West *et al.*, as they used a similar design method to refs. 51 and 52 of the main text upon which we based our design, and that immersion metalens have been previously demonstrated by Ho *et al.* and Chen *et al.* We

appreciate the reviewer pointing out these important references that we had missed initially; they have been added to the main text (now refs 25, 30, and 28, respectively).

In contrast to this previous work, the significance of our demonstration is the application of an immersion metalens to the problem of efficient optical coupling to quantum emitters. This is a timely application of immense technical importance, and we demonstrate that a metalens is particularly well suited for it. While relevant, none of the references mentioned by the reviewer demonstrate the significant result of our paper: collecting single photons from a quantum emitter.

Action Taken: The references suggested by the reviewer have been added, along with a modification to the following sentence in the fifth paragraph of the introduction, which now reads:

“In particular, diffractive optics^{21,22}, high-contrast gratings²³⁻²⁴, and more recently, dielectric metalenses^{21,25-28} of varying designs...”

We added the following sentence to the end of the same paragraph to point out past demonstrations of immersion metalenses:

“When fabricated at a material interface, a metalens can be designed to use the underlying substrate as an immersion medium^{25,28,30} to overcome total internal reflection losses in a similar manner to a SIL; see Sec. I in the supporting Information.”

In addition, the manuscript has several technical issues: 1. The period of the pillar lattice is too large for proper operation of the lens. To avoid excitation of higher order diffraction orders, the lattice period should be smaller than $\text{wavelength}/(n \cdot (1 + \sin(\theta)))$ where n is the refractive index of the substrate and θ is the deflection angle. The deflection angle at the edge of the lens reported in the manuscript is ~35 degrees and the period of the lattice should be smaller than 183 nm. However, the period selected by the authors is 300nm which causes excitation of higher order diffraction orders even close to the center of the lens and reduces the lens efficiency. This effect can be seen in the simulation result shown in Fig. 1a which shows only a small portion of the lens close to the center actually collimates the light emitted by the dipole source.

Our Response: The use of a fixed pitch for this type of design has been well established, as is discussed in the Methods section of the manuscript. We follow the design procedure previously reported in [refs. 51,52 of the main text], and a similar fixed-pitch design approach was used in two of the papers referenced in the reviewer’s previous comment: P. R. West *et al.* [ref. 25 in the updated manuscript] use an identical condition of $\Lambda < \frac{\lambda}{n} < \frac{1550 \text{ nm}}{3.5} < 445 \text{ nm}$ for Si at telecom

wavelengths and W. T. Chen *et al.* [ref. 28 in the updated manuscript] use a related condition of $\Lambda < \frac{\lambda}{2\text{NA}} < \frac{532 \text{ nm}}{2.2} < 240 \text{ nm}$ at visible wavelengths that would result in a larger pitch of $\Lambda < \frac{\lambda}{2\text{NA}} < \frac{532 \text{ nm}}{2.2} < 320 \text{ nm}$ if applied to our structure. The reviewer is correct that this fixed pitch can lead to excitation of higher order diffracted modes at large deflection angles and is only valid for small angles near normal incidence. As a result, the effective aperture of our metalens is actually smaller than its physical diameter. This was discussed in the Methods section of our initial submission, but it is clear from the reviewer's comment that this discussion needs to appear in the main text and in more explicit detail.

Action Taken: In order to direct the reader to the relevant discussion and clarify the limitations on the useable aperture, we have modified the sixth paragraph of the introduction to read:

“Building on these advances, we leverage diamond’s high refractive index to design and fabricate a 27.9 μm -diameter (19.3 μm effective aperture) metalens etched into the surface of a single-crystal substrate...”

We added red circles to figure 2 to denote the usable aperture. We also used 3D-FDTD simulations to confirm the effective aperture, and we added these results to the SI (Fig. S12 and discussion in Sec V.E).

2. The main advantage of metalenses over conventional diffractive lenses, which are typically fabricated in glass and polymers, is their ability in achieving high NA focusing with high efficiency. Although the current manuscript claims (see 2 below) a high numerical aperture for the lens, it does not discuss/characterize the focusing efficiency of the lens. High NA could be readily achieved by patterning a simple Fresnel zone plate on the diamond surface, the main point of using a binary blazed structure is achieving higher efficiency, which has not been reported in the manuscript. The simulated and measured focusing efficiencies (the ratio of the power in the focal spots shown in Figs. 3a and 3c to the optical power incident on the lens aperture) should be reported.

Our Response: We agree with the reviewer’s suggestion and have simulated the focusing efficiency of both the metalens and an equivalent zone plate (Fig. S12). The simulated focusing efficiency is ~50%, similar to the values reported by W. T. Chen *et al.* [ref. 28 in the updated manuscript, <55% shown in Fig. S5 of their manuscript] and better than the values reported by P. R. West *et al.* [ref. 25 in the updated manuscript, 35% shown in table 1 of their manuscript]. Our

metalens significantly outperforms a Fresnel zone plate, which has a focusing efficiency of approximately 25%.

Since this metalens is intended for photon collection rather than light focusing, we have updated figure 3 in the revised manuscript to present the collection efficiency of emission from an NV center at the metalens focus, rather than the focusing efficiency and Strehl ratio suggested by the reviewer.

Action Taken: We added a plot of the simulated focusing efficiency versus wavelength to the supplementary information (Fig. S12), comparing its performance to a zone plate. We performed additional simulations and measurements to quantify the photon coupling efficiency (Sec VE of the SI) and modified Fig.3 in the main text to include these new results.

3. The characterization of the focal spot of the lens is not conclusive. Although the simulation result predicts a small focal spot, the phase errors due to fabrication might lead to a larger focal spot. The actual measured focal spot of the lens is significantly larger than the predicted one. The authors attribute this issue to the aberrations of the microscope objective used to measure the spot, which might be correct. However, comparing the measured result with the convolution of the simulated spot and the PSF of the objective does not provide evidence of the small size of the focal spot. The convolution operation filters out high-frequency components and similarity between two filtered spots does not mean that the original spots were of similar size. The authors can test this by convolving a 5nm and a 700nm spots with the PSF of the microscope and comparing it with the measured spot.

Our Response: We agree with the reviewer that our initial characterization was qualitative. To quantify the metalens NA, we have performed new measurements of the focal spot using incoherent imaging techniques¹, which achieve substantially higher resolution. Employing an iterative deconvolution procedure (the Lucy-Richardson method), we derive the maximum likelihood estimate for the metalens field profile using the point-spread function of the objective derived from confocal PL measurements of isolated single emitters. The uncertainty in this fitting procedure is dominated not by the deconvolution process but rather by the experimental uncertainty in measuring the objective NA. Ultimately, this procedure yields measurements of the metalens NA versus wavelength with substantially reduced uncertainty (Fig. 3c in the main text). At 700nm, the metalens NA is $1.073^{+0.002}_{-0.038}$. The new measurements and corresponding simulations are included in Figs. 3e-g in the main text. To illustrate how well our model matches the measurement, even with no fit parameters, qualitative comparisons of the simulated metalens

focal spot convolved with the microscope intensity PSF as a function of wavelength have also been included in Figs. S10, and S11 of the SI.

Action Taken: We performed higher resolution measurements of the metalens focal spot as a function of wavelength, and modified Fig. 3 in the main text to reflect the updated data and analysis. A discussion of the new measurements, analysis, and fitting procedure is included in the SI (Sec VD).

The authors also have studied the saturation power of an NV center close to the focal spot to offer further evidence of focal spot size. However, the result of this characterization is also unreliable because of the following two issues. First, very large backgrounds (more than 80 % of the collected signal power) have been subtracted from the signals captured from the NV using the objective lens and the metalens. As a result, a small error in the estimation of the background may shew the measurement results significantly. Second, the effect of the polarization/orientation of the NV center has not been considered. The NV might be emitting more power toward the metalens than the objective.

Our Response: The reviewer rightly points out that estimation of focal spot size based on count rates are subject to experimental challenges and uncertainties. In the revised manuscript, we have performed additional measurements to address these concerns and make the measurement of the metalens performance more quantitative.

In our response to the previous comment, we described an improved approach to accurately determine the metalens focal spot size using measurements of transmitted field profiles. We have also performed additional measurements and simulations to quantify the coupling efficiency of single photons emitted by an NV center. Simulations of the coupling efficiency (Fig 3b) account for the alignment of the NV center's optical dipoles relative to the (100) crystal plane on which the metalens is fabricated. We have further calculated the coupling efficiency as a function of the NA of the collection optics.

To compare these simulations with measurements of the photon collection efficiency, we calibrated the transmission efficiency of our collection optics by illuminating from below an aperture with the same physical size as the metalens (30 μm). From this calibration, we can estimate the overall photon collection efficiency using the saturation count rate and a theoretical prediction of the NV center's emission rate. We are confident in the quantification and correction for background fluorescence, since the same background count-rate is used to correct the photon

autocorrelation function (see Fig S15 and section VI in the SI), where deviations would be readily apparent. However, uncertainty in the theoretical emission rate (due to complicated optical dynamics of the NV center and variations due to different materials environments) means that our estimate serves mainly as a consistency check, but the measurements generally agree with the simulations as described in Sec VE of the Supporting Information.

Action Taken: We simulated the metalens coupling efficiency using a quantitative model for the NV emission pattern (see Fig. 3a,b). New measurements of the transmission efficiency in the metalens collection path together with a thorough analysis of the overall photon collection efficiency from a single NV center enable a qualitative comparison with simulations of the metalens coupling efficiency (Sec. VE in the SI).

Overall, the manuscript presents a potentially interesting application of metalenses, but do not offer much of a novel concept nor technique. The lens has not also been properly designed (lattice constant is significantly larger than what it should be), is most probably is less than 50% efficient, and the characterization results are not conclusive.

Our Response: To the best of our knowledge there have been no previous reports on collecting single photons from an individual quantum emitter using an immersion metalens. The impact of this demonstration on the quantum emitter community is clear from the second reviewer's comments. While we agree that the design can be improved, in this initial demonstration we utilized established design procedures for a small-angle functionality, and we have found similar transmission efficiencies to previous reports. We hope the reviewer will reconsider his or her perspective in light of the clear potential for metalenses to enable efficient control of single photon emission from solid materials and efficient coupling into fibers or other low-NA optics.

Reviewer 2:

Since this is a transparent review, I should note that I was an author of several papers on diamond Solid Immersion Lenses (SILs) fabricated using Focussed Ion Beam (FIB), which is a potentially competing technique the authors compare their work to. I have tried my best to weigh the pros and cons of the two techniques fairly.

The authors present a proof of principle demonstration of a dielectric metalens (ML), designed, and fabricated into the surface of diamond through top down clean room processes for efficient and in-principle objective-less collection of fluorescence from diamond colour centres. They claim that the ML exhibits high transmission efficiency and outperforms a traditional free-space objective. They also suggest the structure is good for miniaturization, on chip electronics, coupling to optical cavities, and routing of photons.

I believe the paper represents an interesting application of dielectric ML, and is a novel demonstration in diamond. The fact that it is implemented using standard Ebeam masked dry etching diamond clean-room processing techniques is a definite plus point - in comparison to for example FIB fabrication of solid immersion lens SILs, in which each structure is made individually, making scalability more difficult. The authors present a pleasing amount of detail, with a pretty decent literature review on the design and fabrication processes, and the analysis appears to be valid (subject to a few comments or clarifications detailed below) - so I do feel that other researchers should be able to repeat the work - and if such structures become well used, then this will likely become a good reference resource.

Our Response: We thank the reviewer for their positive assessment of our work. We would like to emphasize that we do not necessarily see the metalens as a replacement for SILs. Our group has previously used SILs and will continue to do so for research purposes since these structures are manifestly compatible with confocal microscopes. Right now, a fully optimized SIL in combination with high-NA collection optics offers better overall collection efficiency than our metalens, and although in principle a metalens can approach similar performance, reaching this limit will require substantial improvements in the design and fabrication. Rather, as pointed out by the reviewer, difficulties in fabricating SILs make the metalens desirable for applications in which scalability and miniaturization are prioritized. To properly address this point for readers, we have performed additional simulations and comparisons of the photon collection efficiencies achieved in various imaging systems including planar diamond surfaces, SILs, and the metalens.

Action Taken: We have dedicated a new section of the Supporting Information (Sec I) to a discussion of the tradeoffs and performance comparisons of SILs and metalenses. We have also clarified the key advantage of the metalens in the second sentence of the Discussion section:

“By integrating the typical objective/SIL combination onto the quantum-emitter host substrate, the metalens has the potential to enable direct fiber coupling of quantum emitters.”

I do have several concerns regarding the claim that the structures outperform traditional free space objectives, which I feel the authors should either clarify - or potentially tone down the claim.

Part of the concern with this claim is that it relies partly on the comparison of numerical aperture ($NA = n \sin \theta$ where n is the immersion material, and θ is the collection cone half angle) between the ML, and the 'traditional objective' - which may not be the ideal way to compare the two systems since it obscures several important details. The ML appears to show a numerical aperture exceeding 1, up to almost 1.2, which at first look is better than a traditional air objective limited by 1 NA, and comparable to standard oil objectives, typically up to 1.4. The authors appear to use a oil immersion objective with adjustable NA from 0.5-1.3, although in the experimental details they appear to use an NA of 0.75 - which suggests they are using the objective at only ~60% of it's achievable NA. If this is the case it is not necessarily surprising that the ML out performs it. In any case the authors should be much more explicit about what they are comparing the NA of the ML to.

The repeated claim of a NA larger than 1 including in the abstract is also potentially slightly misleading, since as the authors note, because the immersion material is diamond, the maximum achievable NA is 2.42, so in reality the performance of the ML in terms of it's light collecting power is comparable to an NA scaled by the diamond material of ~0.4-0.5. This means that although it will outperform a 0.9 NA air objective collecting from diamond (reduced to 0.4 NA by the diamond refractive index), it will likely not outperform a 1.4 NA oil immersion objective collecting from diamond (reduced to 0.6 NA), and certainly will not outperform either of these objectives in combination with a diamond SIL (which eliminates the NA reduction). I would suggest that the inclusion of either estimated collection efficiencies, the NA scaled by the immersion material, or collection cone half angles would make the comparison more easy to understand, and therefore more valid (although once this is done it might be necessary to temper or remove the claim).

Our Response: We thank the reviewer for bringing up this potential source of confusion; we recognize that discussion of NA and photon collection efficiencies using different mediums and collection geometries deserves more attention. We limited the NA of the objective to ~ 0.76 in order to mitigate spherical aberrations when focusing through $\sim 130 \mu\text{m}$ of diamond, so it is true that we are not making a direct comparison between the metalens and an optimized oil-immersion objective in this first demonstration. At the same time, we have performed additional measurements to quantify the metalens NA (see responses to Reviewer #1), which confirms that the effective NA is >1.0 across most of the NV center's emission band. This means that the metalens can outperform any free-space objective in terms of light-collecting power through a planar surface. As the reviewer correctly points out, the NA of a SIL+objective combination can be much higher, up to a theoretical maximum of 2.4. We have added a new discussion of these different collection geometries to the Supporting Information, together with a table listing the NA and collection angle (Table S1).

Action Taken: We removed the phrase "outperforming a traditional free-space objective" from the abstract. Sec.I in the SI together with Fig. S1 and Table S1 were added to address the comments brought up by the reviewer regarding NA and collection angle comparison. We added clarifications in the results and discussion sections to explicitly address why the NA of the oil immersion objective is limited, and to point out that the saturation count rates are not a direct comparison between the optimized performance of both imaging methods.

In terms of the claims to miniaturization, and compatibility to on chip antennas as compared to SILs, I feel like the authors need to be a little more explicit as to the limits of the production of these ML. I note that the lens they present here has a diameter of $28 \mu\text{m}$, for an NV $20 \mu\text{m}$ below the surface. This is a factor of almost 10 larger than the typical NV/SIL combination which has been used before - which then means that the coupling strength of surface mounted antennas are much less - and the danger of cross talk between NV centres is potentially much greater. Do the authors believe that the ML can be pushed to NV centres closer to the surface - allowing smaller structures and closer electronics - or is there a limit based on the achievable size of the diamond pillars which make it up.

Our Response: With regards to miniaturization, we want to reiterate that while SILs only overcome total-internal-reflection, an immersion metasurface acts as both the SIL and the

objective. However, the reviewer brings up a relevant point regarding the size of the structure and the coupling of NV centers to microwave antennas for spin control. In fact, the metalens we have demonstrated is not so different from the SIL devices currently in use by us and others in the NV-center community. The useable diameter of our metalens is $19\ \mu\text{m}$ and its focal depth is $18\ \mu\text{m}$, so we could get the same optical performance by fabricating a metalens next to a wire/antenna for spin control, where the distance between the wire and defect would be $\sim 27\ \mu\text{m}$. Typical SILs have $\sim 8\ \mu\text{m}$ diameter, but in order to use the full NA of an objective these need to be surrounded by trenches with $>20\ \mu\text{m}$ diameter, so the offset to on-chip wires is almost the same. One needs also to account for the orientation of the ac magnetic field relative to the NV-center symmetry axis, and in the (111)-oriented samples used for photon-critical measurements, this coupling is improved for NV centers deeper below the surface.

Due to the detrimental effects of diamond surfaces on the NV center's optical and spin coherence properties, it is beneficial to use defects deep within the bulk of the material. The design that we used is optimized for deep emitters. However, in cases where it may be of interest to address shallower NVs, e.g. to reduce the device footprint or couple directly into small, single-mode fibers, more advanced metasurface designs can be employed. As the reviewer has pointed out, the metasurface fabrication process is much faster and more scalable than for SILs, and offers the potential to design optimized structures for a variety of related applications.

Action Taken: We have modified the first paragraph of the Discussion section in the main text to highlight the potential for packaging and miniaturization using metalens design. In the second discussion paragraph, we explicitly discuss the possibility of employing a design that incorporates diffraction at larger angles for achieving higher numerical aperture or for imaging NVs closer to the surface of the diamond.

Do the authors have any data on how alignment of the ML to the emitter effects their performance?
Also comments on the field of view might be interesting.

Our Response: In our PL imaging experiments, the field of view of the ML collection path is about $1\ \mu\text{m}$ due to the fact that we used fixed, low-NA collection optics to couple the output of the ML to a single-mode fiber. Theoretically, given the ability to adjust the position and alignment of the collection optics or fiber, the field-of-view of the ML should be comparable to the ML diameter. Indeed, the transmission image shown in Fig. 2d of the main text suggests that the field of view is at least $10\ \mu\text{m}$, when the metalens was imaged using higher-NA optics in a separate optical

microscope. This is much larger than the typical field of view and alignment tolerance for SILs. However, it is likely that emitters whose imaging relies on large angle diffraction would exhibit lower efficiencies.

Action Taken: In the new section I of the Supporting Information (p3), we added a paragraph discussing the field of view and alignment tolerance for the metalens, also with comparisons to the relevant expressions for a SIL.

It would also be interesting to comment on the scalability of the ML - for multiple NVs on a single chip. For example if several NV centres were positioned under one ML, or several NVs each under its own ML, how easy is it to separate the collected photons from each NV. Presumably (?) the field of view for the 'objective' achromatic lens which replaces the microscope objective becomes much larger - which could make it possible to see more NV centres on one chip - however, angular separation might become more challenging.

Our Response: We want to emphasize that the achromatic lens pair is not fundamentally necessary, and its main purpose in our experiments is to include a long-pass filter that prevents the 532 nm excitation light from coupling into the collection fiber. The lens pair can be replaced by a commercially available fiber with a long-pass filter fabricated directly on the front facet. Using one of these fibers, the metalens can be directly coupled into a fiber. To couple to multiple NVs on one chip, we could use a fiber array to couple to each metalens-NV pair. For a design as suggested by the reviewer to use a single metasurface to couple to multiple NVs, one could design a structure that shapes emission from physically separated NVs into different spatial modes that could then be collected by a fiber array or comparable optics, presumably at the cost of efficiency and design complexity.

Action Taken: We added the following sentence to the discussion to emphasize that the metalens enables direct coupling to fibers and require no additional optics:

“While achromatic lenses and a free-space long-pass filter were used in our measurement to prevent the pump beam from entering the collection fiber (L1, L2, and LPF in Fig. 4a), the metalens output can be coupled directly into a fiber using a different excitation geometry or using a commercially available multilayer-dielectric-coated fiber tip (available from Omega Optical, Inc., for example).”

Ultimately it would be good to dispassionately discuss whether the performance of such a ML can match that of a SIL with a high NA objective - how far can the NA be pushed towards 2.42..

Our Response: As the reviewer has pointed out, the theoretical maximum NA of an immersion metalens in diamond is 2.4, corresponding to diamond's refractive index, n_d . Metalenses with NA = 0.99 in air have been demonstrated², suggesting that an immersion metalens with NA = $0.99 \cdot n_d = 2.4$ should be achievable with improved design strategies that are optimized for diffraction at wide angles.

In addition to the NA, when comparing to a SIL or any other collection geometry it is crucial also to consider the full collection efficiency of the optical system, including losses due to scattering, absorption, reflections, or misalignment at each optical element. Our new simulations (see Fig. 3) show that the metalens coupling efficiency, i.e., the ratio of photons being collimated into the collection optics to the photons that are emitted into the solid angle of the metalens, is only about 50%. This points to further opportunities to optimize the design in order to take full advantage of the high transmission efficiencies available for metasurfaces. Even as demonstrated, the overall photon collection efficiency of a metalens coupled directly into a fiber could rival the performance of state-of-the-art free-space setups, since the losses associated with compound objectives and other optical elements (mirrors, lenses, etc) are eliminated.

Action Taken: In the main text, the second paragraph of the discussion now highlights the distinction between our metalens and previous high-NA metalens demonstrations, with the potential for further improvements:

“Unlike previous high-NA metalens demonstrations that relied on diffraction far from the optical axis to focus wide angles^{23,28}, the high NA of our metalens is achieved by using diamond as an immersion medium. This implies that optimized design strategies could yield a diamond metalens with an NA substantially larger than the value of 1.07 shown here, potentially with a value approaching the maximum $NA=n_D=2.4$.”

The new section I of the Supporting Information also discusses the achievable NA for immersion metalens devices together with SIL and objective imaging geometries.

In conclusion I believe this is a good paper, demonstrating a novel technique in diamond, with lots of useful detail on the techniques. I have some reservations on the claims made about the

relative performance of the ML, and on the outlook for how the ML will influence the field - however assuming the authors clarify and/or reduce these claims I think it is interesting enough for Nature Communications.

Our Response: We thank the reviewer for all these comments and especially for the suggestion to clarify the relative performance of the metalens as compared to a conventional microscope objective or SIL+objective combination. We have taken steps to do so in the paper. Ultimately, we believe that the potential for designs with high photon collection efficiency directly into an optical fiber, together with the multitude of optical elements that can be achieved using metasurface design make our demonstration a significant and important contribution to the field.

In addition to this concern I have a few more specific comments on the text.

P1 Para1: 'number of photons detected.' small point, but I would say the collection efficiency is maybe better.

Action Taken: We changed “number of photons detected” to “collection efficiency”.

P1, Para3: 'Purcell enhancement' - it might be worth including a reference to the tunable external cavity style structures of for example Jason Smith, David Hunger, etc. (one example might be "Cavity-Enhanced Single-Photon Source Based on the Silicon-Vacancy Center in Diamond")

Our Response: We thank the reviewer for bringing this reference to our attention, and agree that it is a relevant source for the Purcell enhancement discussion.

Action Taken: We have added the suggested reference (now Ref. 14) to the following sentence on page 1, paragraph 2:

“While a number of nanophotonic structures have been investigated for increasing NV emission through Purcell enhancement¹⁰⁻¹⁴,...”.

P2 Para2: 'narrow wavelength range' how narrow - the nv center has a spectrum over several hundred nm (600nm-800nm), while the commonly used excitation wavelength is at 532nm.

Our Response: The statement was meant to refer generally to quantum emitters in high refractive index substrates, as our approach is not limited to the NV center, nor quantum emitters in diamond. In many situations, only the narrow zero-phonon-line emission is useful for experiments. However, we appreciate the ambiguity of this statement in the context of our paper, where we study emission over the NV center's full phonon sideband. In any case, the sentence in question is mostly referring to the imaging performance rather than the bandwidth.

Action Taken: We have removed the phrase “over a narrow wavelength range” from the sentence in question.

Fig 2: The scale bar here is 5um, while in the supplementary information it looks like the same mask is 10um. Is this an error - or is a field of view effect?

Our Response: The scale bar in Fig. 2 is 5um and the scale bar in Fig. S6 is 10µm. These are correct and the differences in scale bars are the result of differences in fields of view.

Action Taken: We have modified Fig. S6 to explicitly state the scale bar value.

P3 para2: 'We characterize the metalens.....'. A semantic point, but personally I feel it is good to separate experimental characterization of real structures, with simulation of theoretical structures - perhaps you could make this clearer.

Our Response: The paragraph in question introduces Fig 3, where we plot the measured data next to the simulation for a direct comparison and to highlight the agreement between our simulated and measured results. We have added “Simulated” and “Measured” labels to the figure to clarify this distinction.

Action Taken: Additional labels in Fig.3e,f of the main text have been added to mark the distinction between simulated and measurement results.

P3 para2: 'we numerically model the microscopes PSF'. Perhaps you could be clearer how you model the PSF. How does the numerical PSF compare with an experimentally determined PSF. Colour scale bars might be useful here too.

Our Response: The NA of our microscope was experimentally verified by fitting to a PL scan of an isolated NV center, using a numerical model of the PSF as a function of NA. The best-fit PSF is plotted in Fig. S8e of the SI. We have convolved the numerically calculated PSF with simulated ML focal spots to demonstrate the clear agreement with our measured data (Fig. S10a-d in the SI).

Action Taken: Calculations of the NA and the microscope PSF have been added to Sec. VD in the SI. Colorbars and axes have been added with labels to Fig. 3e-g to reflect that the image plots are of normalized intensities. We clarified the sentence in the text to emphasize that the numerical model of the objective PSF derives from measurements:

“To enable this accurate comparison of simulation and measurement, we have numerically modeled the microscope's point-spread function using confocal measurements of isolated NV centers (see Methods and Supplementary Information section VD) and deconvolved it from the measured focus spot to reproduce the metalens' transverse and axial field profiles (Fig 3e).”

P3 para3: 'low reflectivity seen qualitatively'. I may be missing something, but I'm not sure how I can qualitatively see anything like this in 2c - it potentially depends a lot on what the contrast is.

Our Response: By “qualitatively” we meant that one can see that the reflection of the metalens is lower than the planar surface, since the planar surface is much brighter than the metalens, but we acknowledge that this statement was ambiguous.

Action Taken: The sentence in question has been removed, having been replaced with a more quantitative discussion of the metalens coupling efficiency (Fig 3b and associated text).

P3 para3: 'low reflectivity..... to be below 12%'. Perhaps the authors should comment and compare with the expected fresnell reflectivity of a non structured diamond/air interface. Do the pillars or the fabrication in general add any extra scattering.

Our Response: We agree that a visual comparison to the Fresnel reflection at 17% would serve to make our points more clear. We have performed additional simulations and the pillars do add to scattering.

Action Taken: We added a dashed line indicating the planar Fresnel reflectance to Fig.S12a, along with calculations of focusing efficiency (Fig.S12c).

Fig 3: a,b, c, It might be good to add colour scale bars. Also to make it clear what the grey bar at the top of the x,z graphs are - presumably this is the pillar length, but good to make it explicit. f: the ripples are suggested to be because of ghosting from an optical element - but I can't see a corresponding characterisation of this effect as promised in the supplemental information. Without this could it be argued that the ripples come from some sort of self interference in the diamond structure?

Our Response: We thank the reviewer for this suggestion, and have modified the figures accordingly. Since Fig 3. now includes quantitative simulations of the metalens coupling efficiency, we have moved the reflectance measurement and simulation (formerly Fig. 3f) to the supplement (now Fig. S12a). With regards to the ripples in the reflectance spectrum, we have simulated the expected spectrum from our metalens which does not exhibit the same ripple pattern as observed in the measured data, leading us to believe that this effect is indeed due to ghosting in the optics used in the reflectance collection path or interference within the diamond sample.

Action Taken: Color bars have been added to Fig. 3e-f, and the figure caption has been modified to make it explicit that the grey bars of the z-scans are indeed the pillar length.

Figure 4: b,c Colour scale bars! f:Would it be possible to make explicit how far $g^2(0)$ goes down. Also - in order for the reader to see how good/bad the background levels are it is nice to add a second axis on the right with the absolute correlations. The left hand scale bar looks a little odd.

Our Response: Since we have performed extensive analysis on the background for this measurement, we have consolidated the data and analysis and plotted it in a separate figure (Fig.S15) in the SI. This figure includes the raw photon correlation data.

Action Taken: Color bars have been added to Fig. 4b,c. An explicit reference to value of $g^2(0)=0.175\pm 0.031$ has been added to the main text.

P8 Para 4. 'Assuming that the excitation and collection paths have similar transmission

efficiencies...' since this is an attempted calculation of the collection efficiency it would be good to explicitly look at the transmission efficiencies. I note that the lenses used in the 4f system are rated with AR coatings from 400-700nm. In my experience there can be much worse performance outside this range - which might end up causing poor performance in the confocal arm (where there are at least 3 extra optical elements). It should be possible to characterize or at least estimate the performance of each arm separately.

Our Response: We agree, more quantitative measurements of the photon collection efficiency are beneficial. Please see also our response to Reviewer #1 on this point.

Action Taken: Transmission efficiencies of the excitation and collection paths have been measured and taken into account for a more quantitative calculation of the photon collection efficiency. This is described in Sec. VE of the Supporting Information.

Reviewer 3:

This is an interesting paper presenting an alternative approach to the collection of PL from single NV centres buried inside diamond. The use of a metalens fabricated inside the diamond as an alternative to a SIL is certainly an idea worthy of exploration.

The paper is very well written and very clear.

However, the results presented fall somewhat short of what was promised in the abstract. Figure 4 certainly shows that the metalens works to collect the light from an NV centre, but this is on the back of the centre being illuminated by the output of an oil immersion objective. Figure 4 would be more interesting if it demonstrated that the ML can actually eliminate the need for an objective lens. Although the increase in the number of collected photons is impressive, I wonder if optimization of the traditional objective arrangement could not have yielded a similar result. Can the experiment be repeated so that the metalens provides the excitation and collection just as the objective lens does in figure 4?

Our Response: We appreciate the reviewers encouraging remarks regarding the novelty and clarity of our manuscript. We also appreciate the reviewer's point regarding the need in this case to use an objective together with the metalens for confocal excitation and collection of PL from a single NV center. The current phase profile design suffers from chromatic aberration such that the focused 532 nm pump beam and PL band collection volume (600 nm to 800 nm) are offset by several microns. A bigger problem, actually, is the use of a single fiber for both excitation and collection, since the 532nm excitation light generates background photons due to Raman scattering in the fiber that overlap with the NV center's PL band. For both of these reasons, it is preferable in the current demonstration to separate the excitation and collection optics.

However, we wish to emphasize that we have designed the structure to *collect single photons* from a quantum emitter for this initial demonstration and that excitation light can be delivered in a variety of ways. For example, the metalens phase profile can be designed using methods outlined in refs. 31-33 in the main text such that it co-focuses the pump beam and collection volume. Alternatively, the objective used in Fig. 4 can be replaced with a second metalens on the backside of the diamond, similar to the approach taken in Brady *et al.*, Appl. Phys. B 103 (2011) and proposed by ref. 20 of the main text where the excitation and collection paths of a trapped ion are routed through separate Fresnel lenses, each designed for the appropriate wavelength. Since these approaches would require the design, fabrication, and measurement of entirely new

devices, we feel that it is outside of the scope of our current work; however we do feel that it is important to make these next steps clear in the discussion section of the paper, as well as accurately convey the scope of our current work in the abstract.

Action Taken: The following sentence in the abstract was changed from “The metalens exhibits high transmission efficiency and a numerical aperture greater than 1.0, outperforming a traditional free-space objective,” to “The metalens exhibits high transmission efficiency and a numerical aperture greater than 1.0, enabling efficient fiber-coupling of quantum emitters,” to more accurately describe the scope of our demonstration.

The following sentence was added to the first paragraph of the Discussion section on pg. 4:

“Going forward, achromatic metalens designs³¹⁻³³ can enable co-focusing of multiple wavelengths, or a second metalens can be incorporated on the backside of the diamond to focus the pump beam³⁴, replacing the objective in our experiment.”

A minor point relates to figure 3f which shows a reflectivity of around 12% or so. It would be helpful to compare this to the measured reflectivity of unmodified diamond surface.

Our Response: We thank the reviewer for their suggestion and agree that a comparison of the metalens reflectivity to the reflectivity of planar air/diamond and immersion oil/diamond surfaces would be useful to the reader. See also our response to a similar suggestion by Reviewer #2. Since we have made changes this Fig.3 in the main text, this specific plot has been moved to the SI (Fig.S12a).

Action Taken: We added a dashed line indicating the planar Fresnel reflectance to Fig.S12a, along with calculations of focusing efficiency (Fig.S12c).

References

¹T. R. Corle and G.S. Kino, *Confocal scanning optical microscopy and related imaging systems*. Academic Press (1996).

²R. Paniagua-Dominguez, Y. F. Yu, E. Khaidarov, S. Choi, V. Leong, R. M. Bakker, X. Liang, Y. H. Fu, V. Valuckas, L. A. Krivitsky, et al., *Nano lett.* **18**, 2124 (2018).

³W. T. Chen, A. Zhu, V. Sanjeev, M. Khorasaninejad, Z. Shi, E. Lee, and F. Capasso, *Nat nano* **13**, 220-226 (2018).

Reviewers' comments:

Reviewer #1 (Remarks to the Author):

Similar to my previous assessment, despite the limited level of novelty, I still believe this might be an interesting application for a metalens, and the authors have now tried to address some of my comments. Nevertheless, there are still several issues with the design and characterization of the metalens, and the authors' interpretation of the results. I have summarized some of these issues below:

1. As I mentioned in my original comments, the use of a lattice constant as large as 300nm for a wavelength of 700nm in a material with an index of 2.4 is simply incorrect, and severely affects the efficiency. To be more clear, even a uniform lattice with a lattice constant of 300nm has higher diffraction orders in diamond, because the lattice should be smaller than $700/2.4 \sim 292$ to be non-diffractive. When designing a lens with an $NA \sim 1$, the lattice constant should be smaller than $700/(2.4+NA) \sim 206$ nm. The problem is not using a single lattice constant over the whole lens area, as the authors seem to be suggesting in their response, but using too large of a lattice constant. The effect of this large lattice constant is seen in Fig. 3e, where the aberrations show a 4-fold rotational symmetry that is stemming from the large lattice. For a proper design, the focus would have circular symmetry. Besides, the fact that a few other groups have made the same mistake doesn't make this design correct. The authors should at least clarify in the text that this large lattice results in unwanted diffraction inside diamond. In addition, since even the uniform lattice is diffractive inside diamond (i.e., it is not subwavelength) they should revise Section II of the SI and plot the transmission and reflection in all existing orders.

2. The authors state that the field of view of the ML is at least 10 μ m. How exactly is this claim made? To be more precise, even looking at the qualitative results of Fig. 2d, one could see that the image is totally blurred outside a diameter of about 5 μ m. With better characterization, the authors would see that the field of view would be limited to about 2-3 degrees (i.e., $\sim 2\mu$ m) because of aberrations. In fact, this has previously been shown by others (see Nat. Commun. 7, 13682, and Ref. 33) of the current work. As an aside, I suggest that the authors be more careful when citing previous works. Ref. 33, which is cited in regards to chromatic aberration concerns is in fact not related to chromatic dispersion at all.

3. On the focal spot measurement issue (previous comment 3), the authors have not really addressed my comment. The agreement between the measured focal spot and the convolved PSF of the objective and ML does not mean that the ML PSF is estimated correctly. When passing through an objective lens with an NA of 0.76, it is almost impossible to retrieve any data with an NA higher than 0.76. This happens simply because any data pertaining to spatial frequencies higher than $NA \sim 0.76$ is completely lost when passing through the objective which effectively acts as a low-pass filter. To see this, I strongly suggest that the authors calculate the convolution of the objective PSF,

with hypothetical PSFs corresponding to NA values ranging from 3 to 0.8, and compare the results with their measured focal spots.

4. The reflectance measurements could be very inaccurate because a large portion of the reflection from the metalens is not within small angles and is therefore not collected by the optics. Although the authors mention this very briefly in SI to justify the lower measured value in comparison with simulations, they should clarify in the main text that this is in fact a lower bound on the reflectance.

5. The exact focusing efficiency definition and calculation method should be clarified. The authors state that the simulations show a 50% focusing efficiency, but it is not clear at all how they have defined and calculated this efficiency from the simulation results. In addition, one can see in Fig. 3a that there is substantial power not collimated within small angles after the ML (i.e., the multiple focuses observed at very near distance after the ML). As a result, the authors should plot Fig. S12b up to much larger angles (close to 90 degrees) to include the substantial amount of power between 50 and 70 degrees (as seen in Fig. 3b).

Reviewer #2 (Remarks to the Author):

As I noted in my previous review, I think this paper is an interesting first demonstration of Meta Lens (ML) used to collect emission from single quantum emitters in diamond. In the previous iteration of the paper I had a few reservations, however after the revisions and additions to the paper, and the rather substantial supplementary information, I think the authors have resolved the major concerns I had and - I think - have improved the paper with extra context and explanation, as well as more detailed justification of how the ML can be used and improved in the future.

I would be happy to recommend the paper for publication in Nature Communications.

Reviewer #3 (Remarks to the Author):

The authors have addressed my comments satisfactorily. The manuscript and supplementary information are very detailed, allowing others to try out this technique.

Itemized responses to the Oct 31st reviewers' comments concerning our manuscript appear below. Referee comments are formatted in blue text. Responses to each comment are in black text along with a summary of the action taken in the manuscript.

Reviewer 1:

Similar to my previous assessment, despite the limited level of novelty, I still believe this might be an interesting application for a metalens, and the authors have now tried to address some of my comments. Nevertheless, there are still several issues with the design and characterization of the metalens, and the authors' interpretation of the results. I have summarized some of these issues below:

Our Response: We appreciate the reviewer's continued interest in our work and their attention to these technical details. In addressing them, we have clarified some aspects of the text and our analysis in ways that will make this paper more useful to other researchers interested in designing metalenses for applications in solid-state quantum photonics.

1. As I mentioned in my original comments, the use of a lattice constant as large as 300nm for a wavelength of 700nm in a material with an index of 2.4 is simply incorrect, and severely affects the efficiency. To be more clear, even a uniform lattice with a lattice constant of 300nm has higher diffraction orders in diamond, because the lattice should be smaller than $700/2.4 \sim 292$ to be non-diffractive. When designing a lens with an $NA \sim 1$, the lattice constant should be smaller than $700/(2.4+NA) \sim 206$ nm. The problem is not using a single lattice constant over the whole lens area, as the authors seem to be suggesting in their response, but using too large of a lattice constant. The effect of this large lattice constant is seen in Fig. 3e, where the aberrations show a 4-fold rotational symmetry that is stemming from the large lattice. For a proper design, the focus would have circular symmetry. Besides, the fact that a few other groups have made the same mistake doesn't make this design correct. The authors should at least clarify in the text that this large lattice results in unwanted diffraction inside diamond. In addition, since even the uniform lattice is diffractive inside diamond (i.e., it is not subwavelength) they should revise Section II of the SI and plot the transmission and reflection in all existing orders.

Our Response:

The reviewer is correct in the assessment that our lattice spacing of 300 nm allows first-order diffraction within the diamond. (This spacing does however satisfy the non-diffraction condition for the free-space side of the metalens). Nevertheless, neither this design parameter, nor the resulting focus spot aberrations, affect the device performance as drastically as the reviewer suggests. To assess the impact of the lattice constant on the metalens coupling efficiency, we designed and simulated a metalens of the same diameter and phase profile but with a lattice constant of 200 nm, which fully satisfies the zero-order condition proposed by the reviewer. The steady-state field intensity and calculated coupling efficiency can be seen in Fig.S13 (recreated below). While the reduced pitch design provides a slight improvement in total transmission – evidenced by a $\sim 6.5\%$ higher coupling efficiency for collection $NA = 1$ in Fig.S13b – this is due to increased forward scattering into large angles that does not improve coupling into low-NA collection optics. The express purpose of our device is to couple light into a small NA collection optic, e.g., for direct fiber coupling. We conclude that our design cannot be improved simply by decreasing the lattice spacing as suggested by the reviewer, a conclusion that is supported by Fig. 4 of ref. 21 of the main text. Indeed, more efficient metalens designs employ alternative

strategies at these large incident angles that specifically leverage diffraction to achieve improved focusing efficiency (ref. 22 and ref. 27 of the main text, for example).

In regards to this calculation, both sets of simulated data are normalized by the effective diameter (~19.3 μm) of our original design for consistency with the results in the rest of the manuscript.

Action taken:

1. We designed and simulated a metalens of smaller lattice constant with the same diameter and Fresnel phase profile. The coupling efficiency with respect to collection solid angle was calculated and plotted in Fig.S13.
2. Details regarding the reduced pitch simulation, including challenges and strategies for further improving the performance of a metalens coupled to a solid-state quantum emitters have been added to Sec. VE of the SI (pg. 20 p.4).

Figure S13. **Simulating a reduced pitch design.** **a**, Steady-state field intensity simulated for a metalens of reduced lattice constant (200 nm). The vertical dashed lines correspond to the effective aperture of the original design. **b**, Coupling efficiency as a function of NA of the collection optic above the metalens for a lattice constant of 300 nm (blue) and 200 nm (red). The acceptance angle, θ_{max} , corresponding to collection NA is plotted as a second x-axis. Dot-dashed and solid black lines indicate NA values of 0.1 and 0.18, corresponding to the NA of the optical fiber and collimating lens used in the measurements, respectively.

2. The authors state that the field of view of the ML is at least 10 μm . How exactly is this claim made? To be more precise, even looking at the qualitative results of Fig. 2d, one could see that the image is totally blurred outside a diameter of about 5 μm . With better characterization, the authors would see that the field of view would be limited to about 2-3 degrees (i.e., ~2 μm) because of aberrations. In fact, this has previously been shown by others (see Nat. Commun. 7, 13682, and Ref. 33) of the current work. As an aside, I suggest that the authors be more careful when citing previous works. Ref. 33, which is cited in regards to chromatic aberration concerns is in fact not related to chromatic dispersion at all.

Our Response: We estimated the field of view based on the size of the image formed through the metalens in Fig. 2d; however, we agree that aberrations degrade the off-axis point-spread function, limiting the field of view to ~2 μm . We did not directly measure the field of view since we are using the metalens to address a point source close to the optical axis, and this is not an important performance metric for our target application. We thank the reviewer for pointing out this mistake and clarified the realistic field of view in section I of the SI.

We thank the reviewer for noticing the mistaken citation; we made an error during the revision process and have corrected it.

Action taken:

1. We have removed Ref. 33 from citations related to chromatic aberrations.
2. We have pointed out the realistic field of view of the device and added Ref. 33 (now Ref. 7) in the SI (pg.3 p.3).

3. On the focal spot measurement issue (previous comment 3), the authors have not really addressed my comment. The agreement between the measured focal spot and the convolved PSF of the objective and ML does not mean that the ML PSF is estimated correctly. When passing through an objective lens with an NA of 0.76, it is almost impossible to retrieve any data with an NA higher than 0.76. This happens simply because any data pertaining to spatial frequencies higher than $NA \sim 0.76$ is completely lost when passing through the objective which effectively acts as a low-pass filter. To see this, I strongly suggest that the authors calculate the convolution of the objective PSF, with hypothetical PSFs corresponding to NA values ranging from 3 to 0.8, and compare the results with their measured focal spots.

Our Response: We appreciate the reviewer's concern regarding the constraints on our analysis of the metalens NA given the limited resolution available to the imaging optics. We agree that we cannot retrieve any data outside of the collection angle of the $NA \approx 0.76$ objective, but we can get a maximum likelihood estimate of the missing information using well established image deconvolution techniques¹. In fact, we can retrieve information about the metalens field profile up to a maximum NA that exceeds the objective NA by a fair margin. The reason for this lies in the improved resolution of the incoherent imaging technique we adopted in the previous round of revisions (see Methods section of the Main Text). Nonetheless, the reviewer's suggestion has motivated a more careful consideration of the experimental uncertainty in this measurement.

To quantify the resolving power of our analysis, we followed the reviewer's suggestion to convolve the objective PSF with hypothetical PSFs with metalens NA values ranging from 0.6 to 2. The results for $\lambda = 700$ nm are shown below in Fig. R1, where we plot the full-width-at-half-maximum (FWHM) for hypothetical field profiles described by both Airy functions and their Gaussian lineshape approximations², as well as the FWHM that results from incoherent convolution with the objective PSF (assumed to be an Airy function). By comparing the FWHM of these hypothetical field profiles to that of the simulated metalens field profile and its incoherent convolution (horizontal dashed lines in Fig. R1), we see that Airy and Gaussian approximations start to diverge from each other – and from the simulated results – upon convolution with the objective PSF. The vertical dashed line signifies the simulated NA, which intersects the convolved FWHM somewhere between the two approximations. Qualitatively, it is understandable that the metalens field profile lies between these two extremes due to aberrations and the radial dependence on coupling efficiency.

Since the incoherent convolution of the Airy disk results in a smaller FWHM for a fixed NA, it serves as a well-constrained lower bound on the estimated NA of our measured focal spots. For each wavelength, the lower bound obtained in this way is consistent with the confidence intervals obtained by fitting the deconvolved field profiles given experimental uncertainty in the objective NA. We have further optimized the iterative Lucy-Richardson deconvolution procedure to yield a deterministic, reversible result by reconvolving the output with the objective PSF and choosing a

number of iterations to minimize the mean-squared error in comparison with the raw data. This re-analysis has produced slight changes to the values and confidence intervals plotted in Fig. 3c and to the deconvolved field profiles in Figs 3f,g, S10, and S11. In all cases, the metalens NA is confirmed to be substantially larger than 0.76.

Action taken:

1. We have performed a systematic study of how the metalens NA is related to the measurable data by convolving known ideal focus field profiles with our measured objective point spread function. This provides a lower bound on the measured NA of our metalens that is consistent with the results of our deconvolution analysis.
2. We improved the deconvolution analysis and recalculated the best-fit metalens NA values and deconvolved field profiles.

Figure R1. **Incoherent convolution analysis.** Full-width-at-half-maximum (FWHM) for hypothetical field profiles described by both Airy functions (red curve) and their Gaussian approximations (blue curve), as well as the FWHM that results from incoherent convolution with the objective PSF (magenta and black curves, respectively). The FWHM of the simulated focal spot (blue dashed line), incoherently convolved simulated focal spot (orange dashed line), and measured focal spot (green dashed line) are plotted as horizontal lines. The simulated NA at 700 nm (purple dashed line) is plotted as a vertical line.

4. The reflectance measurements could be very inaccurate because a large portion of the reflection from the metalens is not within small angles and is therefore not collected by the optics. Although the authors mention this very briefly in SI to justify the lower measured value in comparison with simulations, they should clarify in the main text that this is in fact a lower bound on the reflectance.

Our Response: We thank the reviewer for this suggestion and we have modified the main text to explicitly mention that the figure referenced in the SI represents a lower-bound on reflectance.

Action taken:

1. We have modified the relevant sentence in the first paragraph of Results to read:
“Since the effective refractive index of each pillar is between the refractive index of air and the refractive index of diamond, the metalens is inherently anti-reflective (see simulation and measured lower-bound on reflectance presented in Supplementary Fig.S12), as evidenced by the bright-field reflection microscope image shown in Fig.2c.”

5. The exact focusing efficiency definition and calculation method should be clarified. The authors state that the simulations show a 50% focusing efficiency, but it is not clear at all how they have defined and calculated this efficiency from the simulation results. In addition, one can see in Fig. 3a that there is substantial power not collimated within small angles after the ML (i.e., the multiple focuses observed at very near distance after the ML). As a result, the authors should plot Fig. S12b up to much larger angles (close to 90 degrees) to include the substantial amount of power between 50 and 70 degrees (as seen in Fig. 3b).

Our Response: We appreciate the reviewer’s suggestion and have expanded the discussion of the focusing efficiency calculation. The focusing efficiency represents the power flowing through a small surface located in the focal plane when a planewave is incident on the metalens. We calculate this numerically, using Lumerical, by integrating the z-component of the Poynting vector over a $3\ \mu\text{m} \times 3\ \mu\text{m}$ surface at the metalens focus for each wavelength. Within this surface, the incident fields are entirely dominated by the focal spot (see Fig.3e in the main text). In Fig.S12c, the focusing efficiency is normalized by the power incident on the effective aperture of the metalens, with diameter of about $19.3\ \mu\text{m}$.

We have also adjusted Fig S12b as the reviewer suggests.

Action taken:

1. We added clarification to Sec. VE of the SI for the calculation of the focusing efficiencies of the metalens (SI pg. 18 p.2).
2. We have replotted Fig. S12b up to larger angles and in log-scale.

Reviewer 2:

As I noted in my previous review, I think this paper is an interesting first demonstration of Meta Lens (ML) used to collect emission from single quantum emitters in diamond. In the previous iteration of the paper I had a few reservations, however after the revisions and additions to the paper, and the rather substantial supplementary information, I think the authors have resolved the major concerns I had and - I think - have improved the paper with extra context and explanation, as well as more detailed justification of how the ML can be used and improved in the future.

I would be happy to recommend the paper for publication in Nature Communications.

Our Response: We thank the reviewer for their suggestions and comments which have helped us improve this work.

Reviewer 3:

The authors have addressed my comments satisfactorily. The manuscript and supplementary information are very detailed, allowing others to try out this technique.

Our Response: We thank the reviewer for their suggestions and comments which have helped us improve this work.

List of other changes to the manuscript

1. The order of the lead authors with equal contributions has been swapped.
2. Garrett Kaighn has been added as an author, in recognition of his contributions in performing additional simulations during revisions.

References

- ¹N. Dey, L. Blanc-Feraud, C. Zimmer, P. Roux, Z. Kam, J.-C. Olivo-Marin, and J. Zerubia, Microscopy Research & Tech. **69**, 4 (2006).
- ²B. Zhang, J. Zerubia, and J.-C. Olivo-Marin, Applied Optics **46**, 10 (2007).

REVIEWERS' COMMENTS:

Reviewer #1 (Remarks to the Author):

The authors have satisfactorily addressed my comments.

Reviewer #1 (Remarks to the Author):

The authors have satisfactorily addressed my comments.

Our Response: We thank the reviewer for their suggestions and comments which have helped us improve this work.